# Efficient DAG Learning via Modular Subgraph Integration

## Abstract

Learning causal structures from observational data remains a fundamental yet computationally intensive task, particularly in high-dimensional settings where existing methods face challenges such as the super-exponential growth of the search space and increasing computational demands. To address this, we introduce VISTA (Voting-based Integration of Subgraph Topologies for Acyclicity), a modular framework that decomposes the global causal structure learning problem into local subgraphs based on Markov Blankets. The global integration is achieved through a weighted voting mechanism that penalizes low-support edges via exponential decay, filters unreliable ones with an adaptive threshold, and ensures acyclicity using a Feedback Arc Set (FAS) algorithm. The framework is modular with respect to the choice of directed DAG base learner: it does not depend on the learner's internal objective, parametric form, or optimization procedure, and fully supports parallelization. We also theoretically establish finite-sample error bounds for VISTA, and prove its asymptotic consistency under mild conditions. Extensive experiments on both synthetic and real datasets consistently demonstrate the effectiveness of VISTA, yielding notable improvements in both accuracy and efficiency over a wide range of base learners.

## 1 Introduction

Understanding causal relationships from observational data (Pearl, 2009) is critical across numerous fields such as biology (Petersen et al., 2024), economics (Hünermund & Bareinboim, 2023), and healthcare (Sanchez et al., 2022b). Identifying causal structures enables reliable interventions and scientific insights. A common modeling framework represents the system as a causal graph—a Directed Acyclic Graph (DAG) where nodes are variables and directed edges denote causal links (Spirtes et al., 2000). In practice, large-scale observational datasets further complicate structure recovery, as most existing algorithms struggle to scale efficiently. Constraint-based pipelines (Spirtes et al., 2000; Meek, 2013) must search over large conditioning sets while the number of CI tests grows combinatorially with the size of graph, and finite-sample CI tests become unreliable in high dimensions, so early mistakes can easily propagate to later steps. Score-based learners (Chickering, 2002; Loh & Bühlmann, 2014) optimize over a super-exponential DAG space; practical solvers still require heavy global searches or acyclicity constraints with repeated dense updates, driving time and memory up sharply. These disadvantages make them difficult to perform well in large-scale datasets.

Given the challenges of learning large-scale causal structures, divide-and-conquer strategies have emerged as a natural solution. By decomposing the global graph into smaller, tractable subgraphs, these methods significantly reduce computational complexity, particularly in sparse settings, and facilitate parallel or distributed computation. In addition, aggregating local structures often enhances robustness relative to learning the full graph in a single pass. Early approaches expand neighborhoods from a random node (Gao et al., 2017) or apply hierarchical clustering (Gu & Zhou, 2020). More recent work often partition the variable set into local neighborhoods, such as Markov Blankets, before aggregating them (Dong et al., 2024; Mokhtarian et al., 2021; Tsamardinos et al., 2003; Wu et al., 2023). However, the majority of these "conquer" steps rely on fixed heuristics for merging, such as voting thresholds, edge overlap rules, or manual conflict resolution. While simple, such rule-based schemes lack adaptability to noise and offer limited theoretical guarantees for global consistency. DCILP (Dong et al., 2024) formulates the merging process as an Integer Linear Program (ILP) and introduces solver-based reconciliation. Although this approach benefits from

advances in ILP solvers and distributed optimization, it remains NP-hard and often incurs substantial solver overhead. In practice, even moderate-sized subproblems can lead to high memory usage and long runtimes. Alternatively, recent methods like Shah et al. (2024) retain heuristic-based fusion steps, which are efficient but similarly sensitive to noise and lack theoretical support.

In this paper, we propose VISTA (Voting-based Integration of Subgraph Topologies for Acyclicity), a novel modular framework for large-scale causal discovery. The method proceeds in three main stages. First, for each variable we identify its Markov Blanket, thereby reducing the global problem into tractable local neighborhoods. A base learner is then applied to each neighborhood using the data restricted to that subset of variables, producing local subgraphs. Second, these local subgraphs are aggregated through an adaptive voting mechanism that down-weights low-support edges, suppressing statistical noise and inconsistencies. Finally, the aggregated graph is post-processed with an efficient approximation algorithm that enforces acyclicity while preserving as many high-confidence orientations as possible. We also establish a theoretical result showing that the overall error rate of the procedure is bounded above by that of the subgraph-level aggregation, ensuring soundness of the divide-and-conquer strategy.

Crucially, VISTA is base-learner agnostic within the directed DAG-learning setting and is highly efficient for large-scale structure learning.. It makes no assumptions about the internal design or inductive biases of the base learners, places no restrictions on the choice of Markov Blanket identification algorithm, and imposes no conditions on the underlying data distribution beyond standard faithfulness assumptions. It operates purely on the edge-level outputs of local subgraphs and requires only a one-time aggregation without any additional solver or training overhead. This lightweight design makes VISTA framework readily compatible with any causal discovery method while enabling broad applicability across baselines and full parallelism in the divide phase.

Our key contributions include:

- We propose VISTA, a model-agnostic and modular framework that decomposes global DAG learning into node-centered Markov Blanket subgraphs. It is fully plug-and-play with respect to MB identification and local learners, requiring no identifiability or distributional assumptions on the chosen base learners.

- Our aggregation is lightweight, efficient, and edge-level, performing a one-pass weighted voting instead of relying on expensive global searches or solver-based optimization. We derive finite-sample error bounds and an asymptotic consistency guarantee for this aggregation, which explicitly calibrates errors from imperfect base learners.

- Extensive experiments across diverse graphs and a wide range of base learners demonstrate that VISTA remedies the typical performance drop of base learners, consistently improving robustness and scalability over standalone baselines.

## 2 Related Works

**General Causal Discovery Methods:** Classical algorithms recover Directed Acyclic Graphs (DAGs) by either testing conditional independencies or maximizing a score on the discrete space of graphs. Constraint–based methods (Colombo et al., 2012; Spirtes et al., 2000) iteratively remove edges whose endpoints become independent given bounded-size conditioning sets. Assuming faithfulness, with only observational data, a common result in causal discovery shows that one can only recover the causal graph up to its Markov Equivalence Class (MEC) (Andersson et al., 1997; Verma & Pearl, 2022). Therefore, interventional data is required to learn beyond MEC. Many works propose algorithms that aim to learn the graph with minimal interventional data (Choo et al., 2022; Hauser & Bühlmann, 2012; He & Geng, 2008; Shanmugam et al., 2015; Squires et al., 2020; Zhou et al., 2024). There are also works that characterizes interventional MEC (Hauser & Bühlmann, 2012; Kocaoglu et al., 2019; Jaber et al., 2020; Zhou et al., 2025). Score–based searches (Chickering, 2002; Chickering et al., 2004), evaluate a decomposable metric while heuristically exploring the super-exponential DAG space. Hybrid strategies typified by MMHC (Tsamardinos et al., 2006) first identify each variable's Markov Blanket and then run a restricted greedy search. Although provably sound under the

causal Markov and faithfulness assumptions, all three lines are NP-hard and their run time or memory grows super-polynomially with node count, limiting practical use to $\lesssim 10^2$ variables.

Ordering-based methods constitute a distinct and increasingly influential category. These approaches first attempt to infer a topological ordering of variables and then determine parent sets accordingly. Early examples such as DirectLiNGAM (Shimizu et al., 2011) and RESIT (Peters et al., 2014) exploit non-Gaussianity or additive-noise assumptions to infer edge directions from regression residuals. CAM (Bühlmann & Peters, 2016) extends this idea to nonlinear settings via generalized additive models and greedy order search. More recently, SCORE (Rolland et al., 2022) proposes to identify causal ordering by minimizing the variance of the score function, which has inspired several scalable extensions leveraging score-matching or diffusion-based estimation (Montagna et al., 2023a;b; Sanchez et al., 2022a). These methods achieve promising empirical results on graphs with thousands of nodes, but typically rely on strong functional assumptions and remain sensitive to the precision of the ordering.

Besides, recent years have seen a growing emphasis on continuous and differentiable formulations in causal structure learning, aiming to overcome the combinatorial challenges associated with discrete DAG optimization. NOTEARS (Zheng et al., 2018), DAG-GNN (Yu et al., 2019), GraN-DAG (Lachapelle et al.), and their low-rank or log-det variants (Bello et al., 2022; Fang et al., 2023) convert acyclicity into a smooth penalty and learn graphs via gradient-based methods. Reinforcement learning and meta-learning schemes (Wang et al., 2021; Zhu et al., 2019; Lippe et al., 2021) treat node ordering as a policy and bypass explicit acyclicity constraints. These methods alleviate combinatorial search but still entail an quadratic adjacency parameterization or an cubic matrix exponential, so memory becomes a bottleneck beyond a few hundred nodes. In summary, although continuous optimization and ordering-based heuristics mitigate the need for discrete search, general-purpose methods typically incur quadratic memory overhead or rely on restrictive assumptions, which constrains their applicability to graphs of moderate size.

**Large-Scale Causal Discovery:** To push causal discovery into the high-dimensional regime, researchers have explored sparsity-aware and parallel variants of the above paradigms. Fast Greedy Search (Ramsey et al., 2017) and parallel-PC (Le et al., 2016) cache CI tests for computing. DAGMA (Bello et al., 2022) and NOTEARS-LowRank (Fang et al., 2023) reduce memory usage by factorizing the weight matrix, achieving 5k–10k nodes, while amortized causal discovery (Löwe et al., 2022) shares a latent decoder across samples to scale to massive time-series. Bootstrap and bagging strategies aggregate multiple weak graphs to improve stability without increasing per-run complexity (Wu et al., 2023; Kaiser et al., 2024). However, these scalable algorithms either rely on heavy solvers such as MILP, strong sparsity assumptions, or lack finite-sample guarantees, motivating alternative divide-and-conquer solutions. As a complementary approach, our proposed VISTA framework addresses these challenges through modular subgraph decomposition and lightweight aggregation, while providing finite-sample error control and scalability to graphs with a large scale of nodes.

**Scalable or Modular Structure Learning:** Partition-based approaches decompose the global graph into overlapping neighbourhoods, learn local substructures, and then reconcile conflicts. Early local-to-global techniques grow random neighbourhoods until conditional independence saturates (Gao et al., 2017). Gu & Zhou (2020); Huang et al. (2022); Bevilacqua et al. (2024) apply hierarchical clustering before local search, whereas Shah et al. (2024) first estimates a coarse skeleton and then partitions it to learn subgraphs in parallel. DCILP (Dong et al., 2024) formulates the fusion step as an integer program that guarantees optimal conflict resolution but suffers from MILP infeasibility on dense regions. Recent ensemble methods perform Markov-Blanket bootstrap with majority or confidence-weighted voting (Wu et al., 2023; Ban et al., 2024), yet provide limited theoretical analysis of the aggregated error. For large-scale causal discovery, several local-to-global or fusion-style schemes decompose a graph and then merge the pieces (Margaritis & Thrun, 1999): a top-down CI-driven partition with set-based stitching (Xie & Geng, 2008; Zhang et al., 2022), global fusion over multiple full Bayesian networks (Puerta et al., 2021), a separation–reunion pipeline that repeatedly searches the structure (Liu et al., 2017), a PC-style progressive skeleton requiring iterative bootstraps (Guo et al., 2024), combination of super-structure and exact score-based search (Ng et al., 2021), and DCILP, which formulates reconciliation as an ILP (Dong et al., 2024). However, these methods are typically algorithm-specific rather than modular frameworks; they either assume correct inputs at merging time, depend on heavy global search or solver-based optimization, or perform essentially uncalibrated frequency-based stitching. There also exists a SADA-based or extended model (Cai et al., 2013; 2018; Rahman et al., 2021), but it is

limited to LiNGAM and lacks a calibration process during merging. By contrast, our framework provides a lightweight, calibrated weighted-voting aggregation that down-weights low-support directions and remains compatible with arbitrary base learners.

## 3 Methodology

Let $\boldsymbol{V} = V_1, \ldots, V_n$ be random variables generated by a structural causal model with mutually independent noises $\epsilon_i$:

$$V_i = f_i(\mathrm{Pa}(V_i), \epsilon_i), \qquad \epsilon_i \perp\!\!\!\perp \mathrm{Pa}(V_i),$$

with $\epsilon_1, \ldots, \epsilon_n$ mutually independent. The corresponding directed graph $\mathcal{G} = (\boldsymbol{V}, \boldsymbol{E})$ is defined by $V_i \to V_j \in \boldsymbol{E}$ iff $V_i$ appears in $f_j$, and is assumed to be acyclic. The observational distribution factorizes as $\mathbb{P}(\boldsymbol{V}) = \prod_{i=1}^{n} \mathbb{P}(V_i \mid \mathrm{Pa}(V_i))$. Assuming causal sufficiency, the *Markov Blanket* $\mathrm{MB}(V)$ of a node $V$ is the minimal set that renders $V$ independent of all others given $\mathrm{MB}(V)$; it consists of parents, children, and *spouses* (other parents of the children). Equivalently, $\mathrm{MB}(V)$ *d*-separates $V$ from $\boldsymbol{V} \setminus (\{V\} \cup \mathrm{MB}(V))$. This locality motivates our divide-and-conquer design: by learning $\mathrm{MB}(V)$, causal discovery can be restricted to the induced subgraph $\mathcal{G}[\{V\} \cup \mathrm{MB}(V)]$, substantially reducing search complexity while preserving relevant adjacencies for $V$.

We introduce VISTA (Voting-based Integration of Subgraph Topologies for Acyclicity), a novel modular framework for large-scale DAG learning that is both model-agnostic and efficient. Instead of searching the full graph, VISTA focuses on edge-level evidence: for each node $V$, we form the subgraph induced by $\{V\} \cup \mathrm{MB}(V)$ and run any off-the-shelf local learner, regardless of its parametric form, identifiability assumptions, or internal design. The resulting local predictions are reconciled by a lightweight weighted voting on each ordered pair $(X, Y)$, which calibrates errors from imperfect base learners, and acyclicity is then enforced by a Feedback Arc Set heuristic (Eades et al., 1993). This modular design makes VISTA fully plug-and-play: MB identification and local learning can be tailored to the data regime, while aggregation and acyclicity remain fixed, scalable, and consistent.

**Proposition 3.1** (Coverage of a DAG by Markov-Blanket Subgraphs). *Let $\mathcal{G} = (\boldsymbol{V}, \boldsymbol{E})$ be a DAG. For each $V \in \boldsymbol{V}$, define*

$$\mathcal{G}' = \bigcup_{V_i \in \boldsymbol{V}} \mathcal{G}\left[\{V_i\} \cup \mathrm{MB}(V_i)\right], \tag{1}$$

*where $\mathcal{G}[S]$ denotes the node-induced subgraph of $\mathcal{G}$ on the node set $S$. Then every edge of $\mathcal{G}$ is present in $\mathcal{G}'$, i.e., $\boldsymbol{E} \subseteq \boldsymbol{E}(\mathcal{G}')$.*

*Proof.* Take any edge $(X, Y) \in \boldsymbol{E}$. If $X \to Y$, then $Y$ is a child of $X$ and $X$ is a parent of $Y$, hence $Y \in \mathrm{MB}(X)$ and $X \in \mathrm{MB}(Y)$. Therefore $(X, Y)$ appears in $\mathcal{G}[\{X\} \cup \mathrm{MB}(X)]$ and in $\mathcal{G}[\{Y\} \cup \mathrm{MB}(Y)]$, and thus in the union $\mathcal{G}'$. $\square$

This coverage property is the foundation of VISTA: once MBs and their local subgraphs are correctly identified, no true edge is lost in the decomposition. Importantly, our framework remains agnostic to the specific MB estimator or local learner, that any method suitable for the data distribution can be plugged in. All subsequent aggregation and acyclicity enforcement operate purely at the edge level and rely only on this coverage guarantee. Besides, as shown in Figure 1, the accuracy of MB identification remains relatively stable as the number of nodes increases, whereas the performance of base learners degrades more sharply. Here for a fair comparison, all curves in Figure 1 are evaluated at the skeleton level, with edge directions ignored. We emphasize that the Markov Blanket curve is not intended as a standalone DAG-learning baseline.

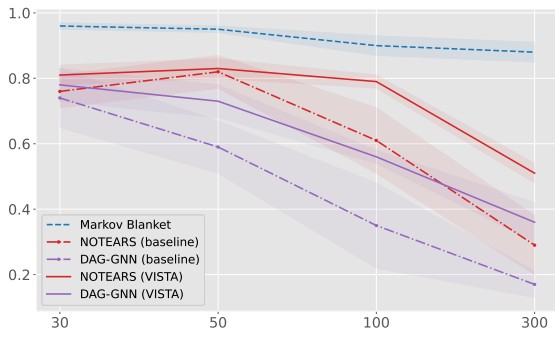

Figure 1: F1 score comparison as the number of nodes increases.

Instead, it measures the quality of the edge-level candidate skeleton induced by the estimated Markov Blankets. This empirical observation is consistent with our theoretical analysis in Section 3.2, where we prove that the proposed merging scheme converges to the correct edge orientations. Furthermore, across different graph sizes, the VISTA-enhanced versions consistently outperform their corresponding baselines, demonstrating the robustness of our framework.

Moreover, since our framework is agnostic to the choice of MB identification methods, we also provide a flexible interface in our implementation that allows practitioners to plug in any suitable MB estimator depending on the specific data distribution. Notably, we assume that each base learner outputs directed edges on local subgraphs throughout this work. Notably, our methods are agnostic with respect to the internal form of directed DAG learners, rather than agnosticism to all possible causal discovery output types.

### 3.1 VISTA: Voting-based Integration of Subgraph Topologies for Acyclicity

**Naive Voting (NV)**  To merge estimated subgraphs into a globally causal graph, we first consider a naive voting strategy. For each pair of nodes $X$ and $Y$, let $A$ denote the number of times the directed edge $X \to Y$ appears across all subgraphs, and $B$ denote the number of times $Y \to X$ appears. The directional support ratio for each orientation is computed as:

$$r_{X \to Y} = \frac{A}{A + B}, \quad r_{Y \to X} = \frac{B}{A + B}.$$

This NV rule serves to demonstrate an important property of our divide-and-conquer framework. The NV decision rule then selects the orientation with larger support: $X \to Y$ is retained if $A > B$, and $Y \to X$ is retained if $B > A$. If $A = B$, the pair is treated as unresolved and no directed edge is added. By Proposition 3.1, every ground-truth causal edge must appear in the union of MB subgraphs. Therefore, even this unweighted scheme, which simply aggregates raw directional votes, already ensures that all true edges are included in the candidate pool. In other words, NV validates that our subgraph decomposition does not lose any causal edges, providing an essential guarantee for the global reconstruction stage.

However, while NV does not distinguish between strong and weak statistical support, edges appearing rarely across subgraphs receive the same confidence as frequently supported ones, and directional conflicts cannot be resolved in a principled manner. These issues motivate the introduction of our weighted voting formulation, which incorporates frequency-based confidence to produce more reliable global orientation decisions.

**Weighted Voting (WV)**  For each pair of nodes $X$ and $Y$, let $A$ and $B$ denote the number of times $X \to Y$ and $Y \to X$ appear across all subgraphs, respectively, and let $m = A + B$ be the total occurrence. We define the confidence-adjusted score as:

$$s(X \to Y) = \left(1 - e^{-\lambda m}\right) \frac{A}{m}, \tag{2}$$

where $\lambda > 0$ is a tunable weighting parameter. An edge $X \to Y$ is retained if $s(X \to Y) \geq t$, where $t \in (0, 1)$ is a global decision threshold. Here, the weighting term $\left(1 - e^{-\lambda m}\right)$ serves as a soft confidence modulator that adapts to the reliability of directional evidence. It plays a role analogous to smoothing priors in Bayesian estimation, where rare events are regularized toward lower confidence. The details in illustrated in Appendix C.1. The inclusion threshold $t$ determines the minimum score required to retain an edge.

Compared to naive voting, which treats all local decisions equally, the weighted scheme jointly calibrates confidence and sparsity. Specifically, the parameter $\lambda$ penalizes edges with weak support, while the threshold $t$ determines the final inclusion criterion. Together, the two parameters govern the

```
def VISTA(nodes, base_learner, ...,
          MB_solver, lam, t):
    local_graphs = []

    for v in nodes:
        MB_v = MB_solver(v)
        G_v  = base_learner(MB_v ∪ v)
        local_graphs.append(G_v)

    G_score = WV(local_graphs, lam)
    G_dag = GreedyFAS(G_score)
    G_final = Filter(G_dag, t)
    return G_final
```

Figure 2: Pseudocode of VISTA framework

precision–recall trade-off, since a larger $\lambda$ tends to preserve edges with limited but consistent evidence and thus improves recall, while a higher $t$ enforces stricter acceptance and thereby improves precision. This mechanism is particularly beneficial in sparse graphs, where many candidate edges receive only minimal support; the exponential weighting amplifies even small differences in frequency, effectively suppressing unreliable edges. As a result, the aggregation remains robust without relying on strong parametric assumptions, and it provides a tunable handle for balancing false discoveries and missed edges. Beyond the divide-and-conquer efficiency of VISTA, the weighted voting strategy itself enhances the performance of base learners, yielding substantial gains in recall while tightening theoretical error bounds. A detailed analysis of these effects is provided in Section 3.2 and Appendices C - D.

**Acyclicity guarantee** While the weighted voting improves robustness, the resulting merged graph may still contain cycles. To ensure that the final output is a valid DAG, it is necessary to explicitly break loops introduced during the merging process. So we explicitly enforce acyclicity by solving a Feedback Arc Set (FAS) problem (Simpson et al., 2016). As FAS is NP-hard, we adopt a fast GreedyFAS heuristic (Eades et al., 1993) adapted to weighted edges; the implementation is detailed in Algorithm 2 in Appendix B.

Notable, GreedyFAS is applied before the final thresholding step in VISTA framework. This ordering is intentional rather than a tunable design choice. The weighted vote graph contains all directional evidence collected from local subgraphs, including low-confidence edges that may still provide useful information about global ordering. Applying thresholding before GreedyFAS would remove part of this evidence before resolving cycles, causing the acyclicity projection to operate on an already sparsified graph. In such a graph, remaining cycles are more likely to consist of relatively high-confidence edges, so resolving them may require discarding stronger orientations that could otherwise have been preserved. Therefore, we first use GreedyFAS to obtain an acyclic projection from the full weighted evidence and then apply thresholding only as a final sparsification step.

In general, our VISTA offers several key advantages that make it particularly suited for large-scale causal discovery. It operates purely on aggregated edge counts and requires only matrix-level operations, with no reliance on optimization solvers or iterative training. Importantly, it is model-agnostic in DAG learning, i.e., the aggregation is independent of the internal structure of base learners and can be applied to any method that outputs directed subgraphs. This modularity allows seamless integration with a broad class of causal discovery algorithms and supports parallel execution in the divide stage. The complete procedure is implemented as a simple and modular pipeline, summarized in Figure 3.

**Theoretical guarantees for Weighted Voting** To ensure the reliability of our edge orientation decisions based on the weighted voting mechanism described above, we provide theoretical guarantees derived from concentration inequalities. The core idea is to determine the minimum number of votes (subgraphs) $m$ required to achieve a desired level of confidence $1 - \epsilon$ in our decision.

**Theorem 3.2** (Sufficient Condition for Weighted Voting Accuracy). *Let $A \sim \text{Binomial}(m, p)$ represent the number of successful votes in $m$ independent subgraphs for the edge direction $X \to Y$, where each subgraph supports this direction independently with probability $p \in (0, 1)$, a decision threshold $t \in (0, 1)$ and the weight function $w(m) = 1 - e^{-\lambda m}, \quad \lambda > 0$. Assume the effective threshold for accepting the edge direction $X \to Y$ is $r(m) = \frac{t}{1 - e^{-\lambda m}} < p$, i.e., the true support rate $p$ is above the effective threshold. Then, if*

$$\frac{mp}{2}\left(1 - \frac{t}{p(1 - e^{-\lambda m})}\right)^2 \geq \log\frac{1}{\epsilon}, \tag{3}$$

*it follows that $P\left(s(A) \geq t\right) \geq 1 - \epsilon$.*

This theorem guarantees that if $m$ is large enough to satisfy the given inequality, the weighted voting procedure will correctly identify the edge direction with high probability. The condition highlights that the required $m$ depends on the squared relative difference between the true probability $p$ and the effective threshold $r(m)$. Note that $r(m)$ itself depends on $m$ and $\lambda$. As $m$ increases or $\lambda$ increases, $1 - e^{-\lambda m}$ approaches 1, and $r(m)$ approaches $t$. The inequality requires larger $m$ and becomes more difficult to satisfy when $p$ is close to $r(m)$ or when higher confidence is desired. This trade-off illustrates the role of $\lambda$ in controlling the conservativeness

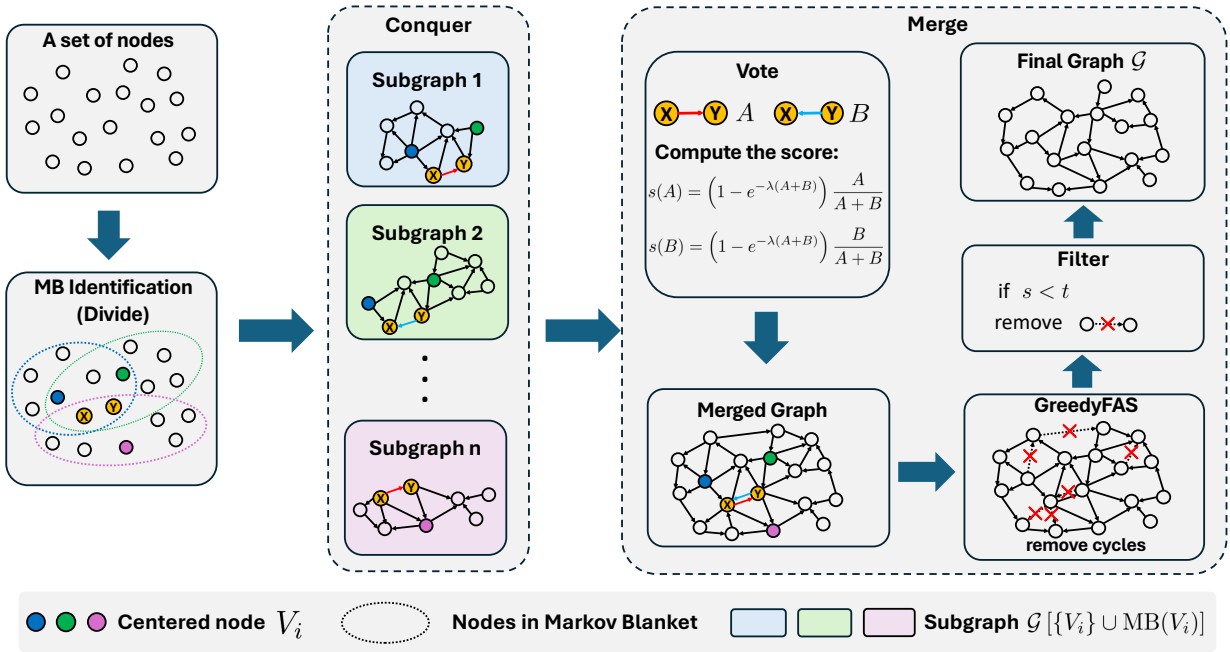

Figure 3: Overview of VISTA, a modular framework for causal discovery: (**Divide**) dividing via Markov Blankets identification, (**Conquer**) parallel subgraph structure identification using a base learner, and (**Merge**) global aggregation through weighted voting. The framework then applies cycle resolution (GreedyFAS) and weight-based filtering to produce the final DAG.

of the decision rule, which we will analyze further in later sections. In practice, the true value of $p$ is unknown, but we can empirically validate the trend predicted by this condition using observed vote frequencies and measured recovery accuracy across different values of $\lambda$ and $t$.

Notably, Theorem 3.2 is stated under an idealized assumption that the votes from different local subgraphs are independent. More concretely, overlap among local subgraphs induces correlations among votes for the same candidate edge. Positive correlation reduces the effective number of independent votes: although an edge may appear in $m$ local subgraphs, the statistical benefit may be closer to that of an effective vote count $m_{\text{eff}} < m$. Therefore, the sufficient condition in Theorem 3.2 should be interpreted as an idealized benchmark or qualitative guide. The same monotone behavior is still expected—larger support, larger separation between $p$ and the effective threshold, and more reliable base learners improve aggregation—but the rate of concentration may be slower when subgraph overlap is strong. Extending the analysis to weakly dependent votes is an important direction for future work.

**Corollary 3.3** (Lower bound on node in subgraphs). *Let $\lambda > 0$, $t \in (0,1)$, and $\epsilon \in (0,1)$ be fixed. For a candidate edge $(X, Y)$, denote by $m$ the number of local subgraphs whose Markov Blankets contain both endpoints. Under the setting of Theorem 3.2, the sufficient condition (3) can be converted into an explicit bound*

$$m \geq \frac{2\log(1/\epsilon)}{p\left((1-t/p)^2 - 2(t/p)(1-t/p)e^{-\lambda}\right)}. \tag{4}$$

Generally, a lower error rate $\epsilon$ leads to a larger $\log(1/\epsilon)$ term, which increases the required size of $m$. When $p$ is much greater than $t$, it results in a small required $m$. This aligns with intuition: if the true voting rate $p$ is far from the threshold $t$, the distinction is easier, and fewer votes are needed for reliable decisions. Similarly, when the gap $p - t$ is small, it will result in a significantly larger required $m$. A large lower bound on $m$ primarily indicates that the current setting yields a very small gap between $p$ and $t$, which, in turn, implies that the decision task has intrinsically high sample complexity.

### 3.2 Error bound analysis

We analyze the edge-level errors of the weighted voting rule to understand how the weighting parameter $\lambda$ and the threshold $t$ affect false positives and false negatives. We first characterize a sufficient condition that converts $t$ into a probability threshold and yields a feasible range for $\lambda$, and then show that under this regime, weighted voting achieves asymptotic consistency as the graph size grows.

**Theorem 3.4** (Practical choice of $\lambda$). *Fix a vote count $m \geq 1$, a decision threshold $t \in (0,1)$, and a target error level $\epsilon \in (0,1)$. If $\lambda$ satisfies*

$$-\frac{1}{m}\ln(1-t) \ < \ \lambda \ \leq \ -\frac{1}{m}\ln\epsilon, \tag{5}$$

*then the weighted-vote rule achieves the prescribed error control under the union bound.*

Theorem 3.4 provides a sufficient theoretical range for $\lambda$ under a fixed vote count $m$, threshold $t$, and confidence parameter $\epsilon$. Here $\epsilon$ should be interpreted as a theoretical failure-probability parameter, analogous to its role in PAC-style bounds, rather than as an experimental hyperparameter. Since the upper endpoint is $-\ln\epsilon/m$, smaller $\epsilon$ values enlarge this endpoint. This reflects the fact that larger $\lambda$ makes the residual damping term $e^{-\lambda m}$ smaller and the confidence weight $1 - e^{-\lambda m}$ closer to 1. While the confidence weight $1 - e^{-\lambda m}$ down-weights low-support orientations at a fixed $t$, the smaller $\lambda$ values impose stricter thresholds $r_\lambda(m)$ to suppress low-support edges, while larger values retain weaker true edges and improve recall. The proof of the theorem, as well as detailed discussions, is in Appendix D.1.

In practice, we adopt the relatively large admissible $\lambda$ in (5), which lowers the effective threshold and reduces false negatives at the cost of more false positives. This choice is well suited to sparse graphs since false positives typically dominate. The empirical behavior of varying $\lambda$ is further examined in Section 4.1. Notably, as $\lambda \to 0$, the rule reduces to naive voting with a fixed threshold $t$. Building on the finite-sample guarantees above, we next analyze the asymptotic behavior of the weighted voting rule as the number of variables grows. Similarly to $p$, let $q \in (0,1)$ denote the probability that a false edge is erroneously included. In practice, both $p$ and $q$ can be empirically estimated.

**Theorem 3.5** (Asymptotic Consistency). *Fix a threshold $t \in (0,1)$ and let $\delta_p = p - t$ and $\delta_q = t - q$ denote the positive margins between $t$ and the inclusion probabilities $p, q$ of true and false edges respectively. Assume $\delta_p, \delta_q > 0$ and that $\lambda$ satisfies the conditions in Theorem 3.4. If every candidate pair receives $m_{ij} \geq m = C \log n$ directional votes with $C > \max\{\frac{1}{2\delta_p^2}, \frac{1}{\delta_q^2}\}$, then we have*

$$\Pr(\text{global error}) = o(1), \qquad \text{as} \ \ n \to \infty. \tag{6}$$

Here, "global error" denotes the event that the final aggregated graph differs from the true graph on at least one candidate edge, that is, at least one true edge is missed or at least one false edge is included after weighted voting. Since most base solvers are reliable and can correctly identify a substantial fraction of true edges, our assumptions are quite mild and practically easy to satisfy. Theorem 3.5 establishes that weighted voting is asymptotically consistent: as the number of subgraph samples increases, the probability of edge-level misclassification vanishes. Notably, the required number of independent subgraphs per edge grows only logarithmically with the graph size $\mathcal{O}(\log n)$, making the approach efficient. From a computational perspective, the global merging procedure involves only one pass of edge counting and scoring, with an overall complexity $\mathcal{O}(n^2)$ regardless of the base learner. These guarantees jointly ensure that the method remains scalable and reliable for large-scale structure discovery. The proof of the theorem is provided in Appendix D.3.

## 4 Experiments

### 4.1 Synthetic data

We empirically evaluate the performance of the proposed VISTA framework on a range of graph structures and sizes, as well as diverse base learners. To demonstrate the improvement and effectiveness of VISTA,

we report representative results that highlight the structural recovery performance of VISTA, its runtime benefits from our modular strategy, and the precision–recall trade-offs induced by different values of $\lambda$. All experiments are conducted on a machine equipped with 13th Gen Intel(R) Core(TM) i9-13900HX CPU (24 cores) and NVIDIA A30 GPU (24GB).

**Baselines** We benchmark VISTA against recent typical state-of-the-art causal discovery algorithms, including NOTEARS (Zheng et al., 2018), DAG-GNN (Yu et al., 2019), and GOLEM (Ng et al., 2020) for the linear setting, which we modeled as linear Structural Equation Model (SEM) with Gaussian noise, as well as SCORE (Rolland et al., 2022) and GraN-DAG (Lachapelle et al.) for the nonlinear setting, defined as quadratic SEM. Each baseline is evaluated both in isolation and when integrated with our modular framework VISTA. Additionally, we provide a comparison between VISTA and DCILP (Dong et al., 2024), a recent distributed framework for causal structure learning, where we also implemented the MB solver used in that work.

We evaluate the accuracy of our VISTA framework under the Naive Voting (NV) and the Weighted Voting (WV) aggregation schemes. Each base learner is tested standalone and with both VISTA variants. We evaluate the proposed method on synthetic datasets generated from Erdős–Rényi (ER) and scale-free (SF) graphs, with average out-degree $h \in \{3, 5\}$ and number of nodes $n \in \{30, 50, 100, 300\}$. Performance is assessed using False Discovery Rate (FDR), True Positive Rate (TPR), Structural Hamming Distance (SHD), and F1 score, as well as runtime metrics. Experiments are conducted under multiple simulation settings, and we report the average performance, with the $\pm$ values indicating the corresponding standard deviations.

**Results** Table 1 shows two complementary roles of our aggregation. The NV variant already lifts recall by pooling evidence from overlapping neighborhoods, recovering more true edges. Building on this, WV acts as a principled edge-level filter. By down-weighting orientations with small or inconsistent support and applying a single global threshold, it removes noisy connections and yields substantially cleaner structures. Quantitatively, WV reduces FDR by $50 \sim 80\%$ relative to the original baselines and by $40 \sim 70\%$ compared to NV, while generally keeping TPR no less than 0.70. The trend holds for both differentiable and combinatorial base learners, indicating that the gains stem from the aggregation rule rather than any particular estimator.

Table 1: Results with linear and nonlinear synthetic datasets ($n = 100$, $h = 5$).

| Method | ER5 | | | | SF5 | | | |
|---|---|---|---|---|---|---|---|---|
| | FDR↓ | TPR↑ | SHD↓ | F1↑ | FDR↓ | TPR↑ | SHD↓ | F1↑ |
| NOTEARS | $0.21 \pm 0.21$ | $0.74 \pm 0.26$ | $208.80 \pm 199.71$ | $0.76 \pm 0.24$ | $0.37 \pm 0.15$ | $0.60 \pm 0.14$ | $352.60 \pm 125.39$ | $0.61 \pm 0.14$ |
| **+VISTA-NV** | $0.87 \pm 0.01$ | $\mathbf{0.97 \pm 0.01}$ | $3171.80 \pm 174.02$ | $0.23 \pm 0.01$ | $0.84 \pm 0.01$ | $\mathbf{0.97 \pm 0.01}$ | $2443.60 \pm 143.74$ | $0.27 \pm 0.01$ |
| **+VISTA-WV** | $\mathbf{0.08 \pm 0.03}$ | $0.68 \pm 0.01$ | $\mathbf{182.40 \pm 16.03}$ | $\mathbf{0.79 \pm 0.02}$ | $\mathbf{0.18 \pm 0.07}$ | $0.68 \pm 0.03$ | $\mathbf{233.00 \pm 34.76}$ | $\mathbf{0.74 \pm 0.03}$ |
| GOLEM | $0.61 \pm 0.16$ | $0.35 \pm 0.17$ | $567.00 \pm 129.77$ | $0.35 \pm 0.15$ | $0.70 \pm 0.15$ | $0.29 \pm 0.19$ | $610.10 \pm 118.00$ | $0.29 \pm 0.17$ |
| **+VISTA-NV** | $0.87 \pm 0.01$ | $\mathbf{0.91 \pm 0.04}$ | $2891.00 \pm 224.42$ | $0.23 \pm 0.01$ | $0.86 \pm 0.01$ | $\mathbf{0.90 \pm 0.02}$ | $2589.00 \pm 270.09$ | $0.25 \pm 0.02$ |
| **+VISTA-WV** | $\mathbf{0.23 \pm 0.12}$ | $0.50 \pm 0.13$ | $\mathbf{306.70 \pm 87.75}$ | $\mathbf{0.60 \pm 0.14}$ | $\mathbf{0.33 \pm 0.15}$ | $0.40 \pm 0.12$ | $\mathbf{371.10 \pm 88.21}$ | $\mathbf{0.50 \pm 0.13}$ |
| DAG-GNN | $0.66 \pm 0.15$ | $0.42 \pm 0.23$ | $739.20 \pm 323.34$ | $0.35 \pm 0.17$ | $0.64 \pm 0.15$ | $0.47 \pm 0.22$ | $731.40 \pm 303.38$ | $0.38 \pm 0.17$ |
| **+VISTA-NV** | $0.87 \pm 0.01$ | $\mathbf{0.95 \pm 0.01}$ | $3065.00 \pm 136.49$ | $0.23 \pm 0.01$ | $0.85 \pm 0.01$ | $\mathbf{0.95 \pm 0.00}$ | $2480.00 \pm 203.65$ | $0.27 \pm 0.01$ |
| **+VISTA-WV** | $\mathbf{0.36 \pm 0.03}$ | $0.56 \pm 0.05$ | $\mathbf{377.00 \pm 26.06}$ | $\mathbf{0.59 \pm 0.02}$ | $\mathbf{0.35 \pm 0.10}$ | $0.49 \pm 0.08$ | $\mathbf{363.00 \pm 41.10}$ | $\mathbf{0.56 \pm 0.09}$ |
| GraN-DAG | $0.92 \pm 0.04$ | $0.05 \pm 0.03$ | $715.00 \pm 70.14$ | $0.06 \pm 0.04$ | $0.94 \pm 0.02$ | $0.05 \pm 0.03$ | $1088.60 \pm 31.49$ | $0.05 \pm 0.02$ |
| **+VISTA-NV** | $0.86 \pm 0.04$ | $\mathbf{0.18 \pm 0.06}$ | $656.60 \pm 83.30$ | $0.16 \pm 0.03$ | $0.89 \pm 0.02$ | $\mathbf{0.20 \pm 0.04}$ | $947.20 \pm 53.33$ | $0.14 \pm 0.02$ |
| **+VISTA-WV** | $\mathbf{0.43 \pm 0.06}$ | $0.10 \pm 0.02$ | $\mathbf{503.40 \pm 46.68}$ | $\mathbf{0.17 \pm 0.03}$ | $\mathbf{0.54 \pm 0.05}$ | $0.11 \pm 0.02$ | $\mathbf{545.80 \pm 65.54}$ | $\mathbf{0.18 \pm 0.03}$ |
| SCORE | $0.92 \pm 0.10$ | $0.58 \pm 0.03$ | $4039.60 \pm 123.3$ | $0.14 \pm 0.15$ | $0.91 \pm 0.03$ | $0.62 \pm 0.05$ | $3166.40 \pm 258.7$ | $0.16 \pm 0.05$ |
| **+VISTA-NV** | $0.95 \pm 0.08$ | $\mathbf{0.76 \pm 0.02}$ | $3464.20 \pm 215.6$ | $0.09 \pm 0.14$ | $0.95 \pm 0.04$ | $\mathbf{0.76 \pm 0.05}$ | $2978.00 \pm 367.3$ | $0.08 \pm 0.07$ |
| **+VISTA-WV** | $\mathbf{0.80 \pm 0.06}$ | $0.65 \pm 0.07$ | $\mathbf{838.00 \pm 364.78}$ | $\mathbf{0.31 \pm 0.09}$ | $\mathbf{0.81 \pm 0.05}$ | $0.63 \pm 0.04$ | $\mathbf{892.60 \pm 345.58}$ | $\mathbf{0.29 \pm 0.06}$ |

Crucially, $\lambda$ appears only in the final aggregation, so sweeping it is retraining-free: we reuse cached votes, recompute $r_\lambda(m)$, and rerun the DAG projection to obtain the full curves. To avoid per-dataset hyperparameter tuning and cherry-picking, all VISTA results in the main tables use a single, fixed operating point: $\lambda = 0.5$ and $t = 0.7$. This choice lies within (5) and serves as a stable compromise between precision and recall across settings. For illustration, if one sets a conventional target confidence $1 - \epsilon = 0.95$, then $-\ln(\epsilon) \approx 3$. Under $\lambda = 0.5$ and $t = 0.7$ gives the admissible interval $(-\ln(0.3)/m, , -\ln(0.05)/m]$, which contains $\lambda = 0.5$ for moderate local vote counts such as $m = 5$. We report the full precision–recall curves, but no post-hoc

selection is performed for the tabulated results. The observed improvement in WV cases against NV aligns with Theorem 3.4. Edges with limited empirical support are selectively pruned while strongly supported ones are preserved, which is exactly the filtering behavior reflected in Table 1. This validates our weighted voting scheme as an effective, model-agnostic mechanism for stabilizing global structures in DAG learning.

Table 2: Results with normalized linear and nonlinear synthetic datasets ($n = 50$, $h = 5$).

| | ER5 | | | | SF5 | | | |
|---|---|---|---|---|---|---|---|---|
| Method | FDR↓ | TPR↑ | SHD↓ | F1↑ | FDR↓ | TPR↑ | SHD↓ | F1↑ |
| NOTEARS | **0.04 ± 0.02** | 0.39 ± 0.01 | 140.00 ± 4.90 | 0.56 ± 0.01 | **0.02 ± 0.02** | 0.38 ± 0.04 | 138.50 ± 9.87 | 0.55 ± 0.05 |
| +VISTA-NV | 0.27 ± 0.05 | **0.61 ± 0.03** | 135.20 ± 6.16 | 0.66 ± 0.02 | 0.35 ± 0.04 | **0.62 ± 0.04** | 132.80 ± 18.82 | 0.63 ± 0.03 |
| +VISTA-WV | 0.19 ± 0.05 | 0.58 ± 0.03 | **122.90 ± 7.54** | **0.68 ± 0.02** | 0.08 ± 0.04 | 0.54 ± 0.06 | **109.10 ± 19.91** | **0.68 ± 0.05** |
| GOLEM | 0.40 ± 0.03 | 0.22 ± 0.04 | 182.00 ± 15.51 | 0.32 ± 0.05 | 0.44 ± 0.07 | 0.20 ± 0.04 | 183.60 ± 6.55 | 0.29 ± 0.05 |
| +VISTA-NV | 0.31 ± 0.03 | **0.75 ± 0.03** | 129.50 ± 4.97 | 0.72 ± 0.02 | 0.29 ± 0.05 | **0.70 ± 0.05** | 122.80 ± 19.87 | 0.70 ± 0.04 |
| +VISTA-WV | **0.06 ± 0.03** | 0.62 ± 0.04 | **95.30 ± 9.88** | **0.75 ± 0.02** | **0.10 ± 0.04** | 0.60 ± 0.06 | **100.20 ± 15.69** | **0.72 ± 0.05** |
| DAG-GNN | 0.16 ± 0.03 | 0.41 ± 0.05 | 160.80 ± 53.55 | 0.55 ± 0.05 | 0.19 ± 0.05 | 0.48 ± 0.04 | 183.60 ± 45.37 | 0.60 ± 0.03 |
| +VISTA-NV | 0.85 ± 0.09 | **0.74 ± 0.14** | 609.80 ± 72.70 | 0.25 ± 0.12 | 0.79 ± 0.04 | **0.72 ± 0.09** | 538.40 ± 25.55 | 0.33 ± 0.05 |
| +VISTA-WV | **0.14 ± 0.05** | 0.50 ± 0.09 | **93.50 ± 29.12** | **0.63 ± 0.07** | **0.13 ± 0.08** | 0.56 ± 0.06 | **87.80 ± 16.56** | **0.68 ± 0.05** |
| GraN-DAG | 0.82 ± 0.01 | 0.06 ± 0.01 | 275.00 ± 18.50 | 0.09 ± 0.01 | 0.92 ± 0.02 | 0.02 ± 0.02 | 269.80 ± 45.50 | 0.03 ± 0.02 |
| +VISTA-NV | 0.66 ± 0.15 | **0.26 ± 0.06** | 219.20 ± 46.41 | 0.29 ± 0.07 | 0.68 ± 0.05 | **0.17 ± 0.04** | 223.00 ± 26.25 | 0.22 ± 0.04 |
| +VISTA-WV | **0.15 ± 0.06** | 0.18 ± 0.05 | **199.20 ± 13.64** | **0.32 ± 0.07** | **0.33 ± 0.03** | 0.13 ± 0.03 | **205.40 ± 59.15** | **0.23 ± 0.04** |
| SCORE | 0.71 ± 0.05 | 0.50 ± 0.05 | 386.80 ± 67.99 | 0.37 ± 0.04 | 0.65 ± 0.13 | 0.52 ± 0.15 | 340.40 ± 81.08 | 0.38 ± 0.05 |
| +VISTA-NV | 0.79 ± 0.03 | **0.60 ± 0.14** | 489.70 ± 123.82 | 0.31 ± 0.04 | 0.77 ± 0.03 | **0.56 ± 0.05** | 471.10 ± 16.68 | 0.33 ± 0.03 |
| +VISTA-WV | **0.64 ± 0.09** | 0.42 ± 0.11 | **305.80 ± 49.93** | **0.39 ± 0.07** | **0.57 ± 0.04** | 0.36 ± 0.06 | **244.20 ± 53.35** | **0.39 ± 0.04** |

To further substantiate this model-agnostic property, we next examine the impact of data standardization as it is known to influence baseline performance (Reisach et al., 2021). The results show that, regardless of fluctuations in the performance of individual base learners, the improvements brought by VISTA remain consistent. This stability further supports our claim that VISTA does not rely on any inductive bias of the base learner or data distribution. Rather, the edge-level aggregation mechanism provides robustness across settings. These findings further highlight the model-agnostic nature of our framework. Additional experiments under alternative parameter settings are provided in Appendix E.3.

We emphasize that VISTA-NV should not be interpreted as the final recommended estimator. Rather, it serves as a diagnostic variant that tests whether the Markov-Blanket decomposition preserves true causal edges in the candidate pool. Since NV aggregates all directional votes without confidence calibration or pruning, it tends to recover many true edges and therefore often achieves high TPR, but it can also introduce a large number of spurious edges, leading to substantially increased FDR and SHD. In contrast, VISTA-WV is the intended aggregation rule. By down-weighting low-support orientations and applying a global threshold, WV removes many unreliable edges introduced by NV. This filtering substantially reduces FDR and SHD, but it can also prune weakly supported true edges, leading to a lower TPR for some base learners. Thus, the main benefit of WV is not uniformly increasing every metric, but providing a more favorable precision–recall trade-off and substantially cleaner global structures. This also explains why WV often improves F1 and SHD even when its TPR is lower than that of NV.

**Time efficiency** To assess the scalability of our framework, we report the total computation time for different base learners in Table 3. All results are presented as mean ± standard deviation over repeated runs. All reported runtimes are measured as end-to-end wall-clock time on the same hardware platform. For VISTA, the local Markov Blanket subgraph learning tasks are executed in parallel using different CPU workers, while the aggregation and acyclicity post-processing

Table 3: Comparison of computing time (s) under ER3 setting.

| Method | $n = 50$ | $n = 100$ | $n = 300$ |
|---|---|---|---|
| NOTEARS | 494.40 ± 98.24 | 1473.69 ± 395.59 | 12515.63 ± 1599.06 |
| +VISTA | **189.15 ± 65.37** | **339.90 ± 158.75** | **2136.72 ± 708.15** |
| GOLEM | 72.65 ± 15.41 | 108.82 ± 70.56 | 261.84 ± 30.44 |
| +VISTA | **21.93 ± 0.81** | **26.16 ± 2.68** | **43.40 ± 3.21** |
| DAG-GNN | 628.63 ± 55.29 | 2192.97 ± 323.59 | 17713.84 ± 2861.06 |
| +VISTA | **201.31 ± 43.36** | **371.25 ± 199.91** | **1960.43 ± 794.02** |
| GraN-DAG | 730.42 ± 89.95 | 3035.76 ± 481.85 | 25205.64 ± 2098.85 |
| +VISTA | **238.53 ± 51.36** | **472.30 ± 172.77** | **2336.32 ± 1028.04** |
| SCORE | 426.63 ± 61.15 | 10040.65 ± 209.31 | ——— |
| +VISTA | **105.64 ± 39.65** | **198.82 ± 34.12** | **225.16 ± 11.45** |

steps are run after all local jobs finish. Across all tested graph sizes, integrating VISTA consistently yields substantial runtime reductions compared to the original methods. These improvements are not due to algorithm-specific acceleration but result directly from our divide-and-conquer design: since each local subgraph is processed independently, the learning procedure naturally supports parallel execution. This decomposition effectively reduces the per-task computational load and alleviates memory bottlenecks, enabling scalable causal discovery even with large node counts. Further results for other settings are included in Appendix E.2.

We sweep $\lambda$ and plot precision/recall in Figure 4. By the conclusion of Theorem 3.4 and Appendix D.1, larger $\lambda$ shifts the method toward higher recall and lower precision by relaxing the penalty on low-support edges. Within the theoretical range, this precision–recall trade-off is smooth and yields informative voting thresholds $r_\lambda(m)$. The figure also substantiate this point, Small $\lambda$ strongly discounts low-support edges, yielding high precision and low recall. Similarly, as $\lambda$ increases, recall rises while precision falls. Beyond the upper end of (5) we have $(1 - e^{-\lambda m}) \approx 1$ and thus $s(X \to Y) \approx A/m$, so the curves plateau and further increases of $\lambda$ have negligible effect. Therefore, to balance precision and recall in practice, a moderate value of the hyperparameter could be fixed within the theoretical range, which serves as a stable operating point.

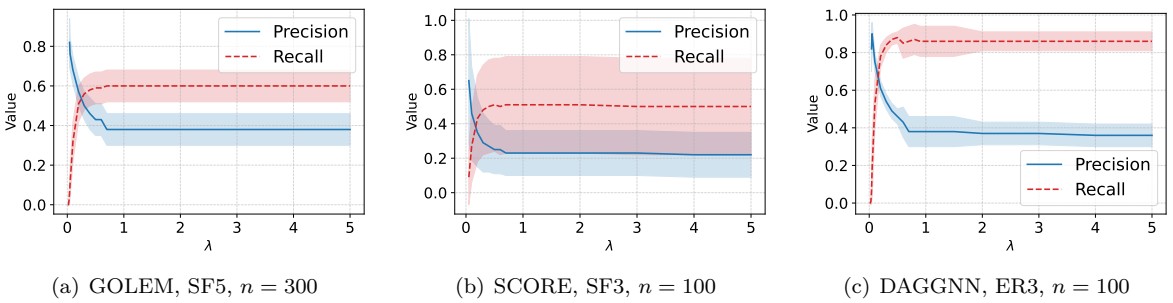

(a) GOLEM, SF5, $n = 300$  (b) SCORE, SF3, $n = 100$  (c) DAGGNN, ER3, $n = 100$

Figure 4: Precision–recall trade-off under varying $\lambda$, where threshold $t = 0.5$.

**Sensitivity study of $\lambda$**

## 4.2 Comparison with DCILP

We provide a detailed comparison between VISTA and DCILP, two methods that share a high-level divide-and-conquer strategy based on Markov Blanket decomposition. Although both approaches follow a similar decomposition principle, they differ notably in how they perform the aggregation step and enforce global acyclicity.

DCILP formulates the merging process as an integer linear program that guarantees the removal of 2-cycles, but relies on iterative post-processing to eliminate larger cycles. This procedure can be computationally intensive and may not always yield globally consistent solutions without additional refinement. In contrast, VISTA enforces acyclicity using a feedback arc set–based heuristic, which is algorithmically simpler and ensures a valid DAG by construction. Another distinction lies in how the two frameworks handle local estimation errors: DCILP applies aggressive pruning to Markov Blanket outputs before global optimization, which may propagate early-stage errors. VISTA instead retains a broader set of subgraph information and applies confidence-aware filtering during aggregation, providing more flexibility and robustness to local variability.

For empirical evaluation, we followed DCILP's implementation baseline by using DAGMA (Bello et al., 2022) as the phase-2 solver in both frameworks. This matched setup enables a controlled comparison under consistent base learners and subgraph configurations.

Results in Table 4 show that, under the same configuration using DAGMA as the local structure learner, both VISTA variants (NV and WV) consistently outperform DCILP across all benchmark settings. Even

Table 4: Comparison of DCILP and VISTA under DAGMA baseline.

| Scenario | Model | FDR↓ | TPR↑ | SHD↓ | F1↑ |
|---|---|---|---|---|---|
| ER5, $n = 30$ | DCILP | $0.74 \pm 0.04$ | $0.52 \pm 0.06$ | $227.00 \pm 27.17$ | $0.35 \pm 0.04$ |
| | VISTA-NV | $0.63 \pm 0.02$ | $\mathbf{0.98 \pm 0.01}$ | $236.80 \pm 14.86$ | $0.54 \pm 0.02$ |
| | VISTA-WV | $\mathbf{0.09 \pm 0.07}$ | $0.75 \pm 0.11$ | $\mathbf{45.80 \pm 23.57}$ | $\mathbf{0.82 \pm 0.09}$ |
| SF5, $n = 30$ | DCILP | $0.81 \pm 0.04$ | $0.49 \pm 0.07$ | $309.70 \pm 60.87$ | $0.27 \pm 0.04$ |
| | VISTA-NV | $0.63 \pm 0.01$ | $\mathbf{0.97 \pm 0.01}$ | $208.20 \pm 7.98$ | $0.54 \pm 0.01$ |
| | VISTA-WV | $\mathbf{0.13 \pm 0.09}$ | $0.85 \pm 0.06$ | $\mathbf{35.00 \pm 16.97}$ | $\mathbf{0.86 \pm 0.07}$ |
| ER3, $n = 50$ | DCILP | $0.79 \pm 0.02$ | $0.49 \pm 0.05$ | $282.50 \pm 26.23$ | $0.29 \pm 0.02$ |
| | VISTA-NV | $0.74 \pm 0.02$ | $\mathbf{0.97 \pm 0.02}$ | $397.20 \pm 41.38$ | $0.40 \pm 0.03$ |
| | VISTA-WV | $\mathbf{0.06 \pm 0.01}$ | $0.76 \pm 0.04$ | $\mathbf{39.00 \pm 3.22}$ | $\mathbf{0.84 \pm 0.02}$ |
| SF3, $n = 50$ | DCILP | $0.91 \pm 0.01$ | $0.52 \pm 0.03$ | $820.40 \pm 110.23$ | $0.15 \pm 0.01$ |
| | VISTA-NV | $0.71 \pm 0.05$ | $\mathbf{0.95 \pm 0.02}$ | $345.00 \pm 71.05$ | $0.44 \pm 0.05$ |
| | VISTA-WV | $\mathbf{0.14 \pm 0.04}$ | $0.84 \pm 0.06$ | $\mathbf{40.80 \pm 12.38}$ | $\mathbf{0.81 \pm 0.08}$ |
| ER5, $n = 50$ | DCILP | $0.80 \pm 0.01$ | $0.52 \pm 0.04$ | $520.20 \pm 29.51$ | $0.29 \pm 0.01$ |
| | VISTA-NV | $0.76 \pm 0.01$ | $\mathbf{0.98 \pm 0.01}$ | $730.80 \pm 24.85$ | $0.38 \pm 0.01$ |
| | VISTA-WV | $\mathbf{0.09 \pm 0.03}$ | $0.83 \pm 0.03$ | $\mathbf{59.20 \pm 10.32}$ | $\mathbf{0.86 \pm 0.02}$ |
| SF5, $n = 50$ | DCILP | $0.90 \pm 0.01$ | $0.49 \pm 0.03$ | $1019.90 \pm 57.87$ | $0.17 \pm 0.01$ |
| | VISTA-NV | $0.75 \pm 0.01$ | $\mathbf{0.97 \pm 0.01}$ | $665.50 \pm 42.65$ | $0.40 \pm 0.02$ |
| | VISTA-WV | $\mathbf{0.10 \pm 0.02}$ | $0.80 \pm 0.02$ | $\mathbf{64.50 \pm 6.50}$ | $\mathbf{0.85 \pm 0.02}$ |

the naive voting variant achieves lower FDR and SHD while maintaining competitive or higher TPR and F1 scores, suggesting that the ILP-based merging step in DCILP may introduce additional overhead without proportional accuracy gains. The weighted voting variant further improves performance by adaptively resolving directional conflicts based on edge support. We also note that as graph size increases such as $n = 100$, DCILP occasionally encounters solver infeasibility or produces solutions with substantially higher error rates, likely due to the combinatorial complexity of ILP formulation. In contrast, VISTA maintains stable performance with reduced computational demands. These comparisons underscore the scalability and robustness benefits of our framework in large-graph causal discovery settings.

## 4.3 Real data

We further evaluate all methods on the well-known Sachs protein signaling network based on expression levels of proteins and phospholipids (Sachs et al., 2005). This benchmark is widely used in causal discovery research, and the ground-truth graph with 11 nodes and 17 directed edges is consistently accepted by the community.

Here we trained normalized data with 853 samples and reported the results in Table 5. Incorporating VISTA consistently reduces false discoveries and improves structural accuracy, measured by SHD and SID (Peters & Bühlmann, 2015) across different baselines. This highlights that VISTA is a plug-and-play module that can reliably enhance the performance of arbitrary causal discovery algorithms.

Table 5: Results on the Sachs protein-signaling network.

| Method | FDR↓ | TPR↑ | SHD↓ | SID↓ |
|---|---|---|---|---|
| GOLEM | 0.80 | 0.26 | 16 | 50 |
| +**VISTA** | 0.57 | 0.18 | 16 | 48 |
| SCORE | 0.81 | 0.18 | 18 | 57 |
| +**VISTA** | 0.60 | 0.12 | 15 | 53 |
| DAG-GNN | 0.50 | 0.12 | 15 | 54 |
| +**VISTA** | 0.25 | 0.18 | 14 | 52 |
| GraN-DAG | 0.82 | 0.53 | 16 | 48 |
| +**VISTA** | 0.00 | 0.29 | 12 | 45 |

To complement the synthetic studies and the small Sachs network with a larger evaluation, we additionally assess VISTA on causalAssembly (Göbler et al., 2024), a semi-synthetic benchmark derived from a real automotive manufacturing line. The results are reported in Table 6. Its ground-truth DAG is established from the documented physics of the production processes and contains 98 variables organized into five sequential stations with 485 directed edges; the accompanying observational data are produced by distributional random forests fitted to the real line measurements so as to strictly adhere to this causal model. We use the observational dataset shipped with the benchmark, drawing samples and standardizing each variable to unit variance, following the recommendation of Reisach et al. (2021) to avoid var-sortability artifacts on continuous data. As in our synthetic experiments, we evaluate the same set of base learners, each in isolation and integrated into VISTA. Markov Blankets are identified with the same estimator and configuration used throughout the paper, and the aggregation uses the single fixed operating point $\lambda = 0.5$ and $t = 0.7$ adopted in all of our main tables, without any per-dataset tuning.

Table 6: `CausalAssembly` experimental results.

| Method | FDR↓ | TPR↑ | SHD↓ | F1↑ |
|---|---|---|---|---|
| GOLEM | 0.82 | 0.02 | 552 | 0.04 |
| **+VISTA** | 0.38 | 0.05 | 477 | 0.09 |
| SCORE | 0.96 | 0.32 | 580 | 0.07 |
| **+VISTA** | 0.62 | 0.21 | 491 | 0.27 |
| DAG-GNN | 0.68 | 0.05 | 480 | 0.08 |
| **+VISTA** | 0.42 | 0.06 | 456 | 0.10 |
| NOTEARS | 0.75 | 0.00 | 472 | 0.01 |
| **+VISTA** | 0.58 | 0.05 | 450 | 0.08 |

## 5 Conclusion

In this paper, we introduced VISTA, a scalable and model-agnostic framework for causal discovery that decomposes global structure learning into Markov Blanket neighborhoods, aggregates them via a weighted voting scheme, and enforces acyclicity through FAS post-processing. The design is fully parallelizable, and the aggregation step operates only at the edge level, enabling efficient exploration of operating points regardless of the base learner. Theoretically, we establish finite-sample error guarantees and asymptotic consistency under mild conditions. Empirically, across diverse graph families and base learners, VISTA improves accuracy and runtime efficiency, typically increasing precision without sacrificing recall.

Despite the favorable performance of VISTA, the framework has several limitations. First, when aggregating local graphs, latent confounding introduced by restricting the learner to subsets may produce high-confidence redundant edges. In some cases these edges do not necessarily participate in cycles and our current framework can only mitigate them through the combination of GreedyFAS and threshold-based filtering. Moreover, although the FAS projection guarantees acyclicity, it may also prune edges that are weakly supported yet correct, which can negatively affect downstream tasks that are sensitive to edge directions. Future work includes incorporating interventional data to improve orientation accuracy and extending the VISTA framework to online settings for large-scale applications.

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

## A  Algorithm of VISTA

## B  Pseudocode of Feedback Arc Set

Given the weighted directed graph produced by the voting stage, we enforce acyclicity by casting the problem as a Feedback Arc Set (FAS) optimization. Since exact FAS is NP-hard, we adopt a greedy approximation based on node degree imbalance. For each node $V_i \in \boldsymbol{V}$, let $d^o(V_i)$ and $d^i(V_i)$ be its out- and in-degrees, and define imbalance $\delta(V_i) = d^o(V_i) - d^i(V_i)$. At each iteration, we remove one node: sources are appended to a sequence $s_1$, sinks are prepended to a sequence $s_2$, and if neither exists, we select the node with the largest absolute imbalance $|\delta(V_i)|$. This process continues until all nodes are removed, yielding a topological order $s = s_1 // s_2$.

---

**Algorithm 1** VISTA-Weighted Voting

---

**Require:** A set of local subgraphs $\{\mathcal{G}_V : V \in \boldsymbol{V}\}$, each induced by the Markov Blanket of node $V$; hyperparameters $\lambda$ and threshold $t$.
 1: Initialize zero matrix `EdgeCount` to record counts for each edge $V_i \to V_j$ where $V_i, V_j \in \boldsymbol{V}$.
 2: **for** each local subgraph $\mathcal{G}_V$ **do**
 3:     **for** each directed edge $V_i \to V_j$ in $\mathcal{G}_V$ **with** $i \neq j$ **do**
 4:       Increment `EdgeCount`$[V_i, V_j]$ by 1.
 5:     **end for**
 6: **end for**
 7: Compute the `Occurrence` matrix as `Occurrence` $\leftarrow$ `EdgeCount` $+$ `EdgeCount`$^\top$
 8: Compute the coefficient matrix elementwise: `Coef` $\leftarrow 1 - \exp(-\lambda \cdot$ `Occurrence`$)$.
 9: Compute the merged weighted directed graph $\mathcal{G}_1 =$ `Coef` $\odot$ `EdgeCount`$/$`Occurrence`.
10: Use Algorithm 2 to break cycles in $\mathcal{G}_1$ and obtain a DAG $\mathcal{G}_2$.
11: Remove edges in $\mathcal{G}_2$ whose weights are less than threshold $t$ to obtain the final DAG $\mathcal{G}$.
12: **return** the global causal graph $\mathcal{G}$.

---

Using the resulting order $s$, any edge $(V_i, V_j) \in \boldsymbol{E}$ that points from a later node to an earlier node in $s$ is marked as a backward edge. These are sorted by weight and the lightest ones are iteratively removed until the graph becomes acyclic. Algorithm 2 summarizes the procedure.

---

**Algorithm 2** Solve FAS to guarantee acyclicity on the weighted directed graph

---

**Require:** A weighted directed graph $\mathcal{G} = (\boldsymbol{V}, \boldsymbol{E})$, where $E_{UV}$ denotes the edge from $U \in \boldsymbol{V}$ to $V \in \boldsymbol{V}$ and $w_{UV} > 0$ is its weight.
 1: For the ease of description, $\mathcal{G}'$ is a copy of input graph $\mathcal{G}$.
 2: Initialize two empty sequences $s_1 \leftarrow \emptyset, s_2 \leftarrow \emptyset$, and a backward edge set $b \leftarrow \emptyset$.
 3: **while** $\mathcal{G} \neq \emptyset$ **do**
 4:     **if** $\mathcal{G}$ contains a source (or a sink) **then**
 5:       choose $U$ as the source (sink) with maximum (minimum) $\delta(\cdot)$
 6:       $s_1 \leftarrow s_1 // U \quad (s_2 \leftarrow U // s_2)$
 7:       $\boldsymbol{V} \leftarrow \boldsymbol{V} \setminus \{U\}$; $\boldsymbol{E} \leftarrow \boldsymbol{E} \setminus \{E_{UX}, E_{XU} \mid X \in \boldsymbol{V}\}$
 8:       $\mathcal{G} \leftarrow (\boldsymbol{V}, \boldsymbol{E})$
 9:     **end if**
10: **end while**
11: The topological ordering is $s = s_1 // s_2$
12: **for** $E_{UV}$ in the input graph $\mathcal{G}'$ **do**
13:     **if** $U$ is after $V$ in $s$ **then**
14:       $b \leftarrow b // E_{UV}$
15:     **end if**
16: **end for**
17: Sort $b$ in ascending order according to $w_{UV}$
18: **while** $\mathcal{G}'$ is not a DAG **do**
19:     remove the edge with smallest $w_{UV}$ from $\mathcal{G}'$
20: **end while**
21: **return** The directed acyclic causal graph $\mathcal{G}'$.

---

## C   Statistical Accuracy Analysis of Weighted Voting

This section provides a theoretical analysis of the statistical behavior of the weighted voting mechanism introduced in Section 3.1. The goal is to characterize the conditions under which a candidate edge is correctly retained or excluded based on its empirical directional support. The analysis builds on a probabilistic

interpretation of the weighted score as a posterior expectation, and derives sufficient conditions for edge-level accuracy using concentration inequalities.

We begin by examining the relationship between the weighting parameter $\lambda$, the empirical support rate, and the effective threshold. We then establish a general bound on the probability of edge-level error under the weighted voting rule, and provide sufficient conditions under specific support distributions that guarantee accurate recovery.

## C.1 Bayesian Motivation for the Weighted Voting Rule

Specifically, we show that the score can be viewed as the posterior mean under a Beta prior whose influence diminishes as the number of supporting subgraphs increases. We first consider each edge direction $X \to Y$ as a binary decision problem. Suppose each local subgraph that includes both $X$ and $Y$ independently votes for one of the two directions: $X \to Y$ or $Y \to X$. Let $A$ and $B$ denote the number of times each direction appears, and let $m = A + B$ be the total number of subgraphs providing directional evidence.

A natural approach is to model the true support probability $p = \Pr(X \to Y)$ using a Beta prior:

$$p \sim \text{Beta}(\alpha, \beta),$$

so that the posterior mean becomes

$$\mathbb{E}[p \mid A] = \frac{A + \alpha}{m + \alpha + \beta}. \tag{7}$$

In classical Laplace smoothing, a fixed prior such as $\text{Beta}(1,1)$ adds uniform pseudo-counts regardless of sample size. However, in our setting, most candidate edges are supported by very few subgraphs. The fixed priors are therefore either too weak to suppress noise or too strong to allow learning when evidence grows.

We therefore introduce a data-dependent pseudo-count that decreases with $m$. Specifically, we set $\alpha = 0$, and define an effective prior strength:

$$\beta = \kappa(m) := \frac{me^{-\lambda m}}{1 - e^{-\lambda m}}, \tag{8}$$

where $\lambda > 0$ is a tunable parameter. This yields the posterior mean:

$$\mathbb{E}[p \mid A] = \frac{A}{m + \kappa(m)} = \left(1 - e^{-\lambda m}\right) \cdot \frac{A}{m}. \tag{9}$$

Thus, our weighted score function $s(X \to Y)$ can be viewed as the posterior mean under a Beta prior whose strength vanishes exponentially as the number of supporting subgraphs increases. When $m$ is small, the exponential decay is slow, and the prior contributes a significant regularization, effectively suppressing low-support edges. As $m$ grows, the prior influence rapidly vanishes, and the score approaches the empirical frequency $A/m$, recovering naive voting.

The hyperparameter $\lambda$ controls how quickly the prior decays. A larger $\lambda$ yields more aggressive penalization for rare edges, while a smaller $\lambda$ allows quicker adaptation to the empirical signal. This dynamic pseudo-count interpretation explains the design of our exponential weight $1 - e^{-\lambda m}$ and its effectiveness in controlling false positives in sparse and noisy settings.

## C.2 Proof of Theorem 3.2

**Theorem 3.2** (Sufficient Condition for Weighted Voting Accuracy) *Let $A \sim \text{Binomial}(m, p)$ represent the number of successful votes in $m$ independent subgraphs for the edge direction $X_1 \to X_2$, where each subgraph supports this direction independently with probability $p \in (0, 1)$, decision threshold $t \in (0, 1)$ and the weight function $w(m) = 1 - e^{-\lambda m}$, $\lambda > 0$. Assume the effective threshold for accept the edge direction $X_1 \to X_2$ is $r(m) = \frac{t}{1 - e^{-\lambda m}} < p$, i.e., the true support rate $p$ is above the effective threshold. Then, if*

$$\frac{mp}{2} \left(1 - \frac{t}{p(1 - e^{-\lambda m})}\right)^2 \geq \log \frac{1}{\epsilon},$$

*it follows that $P\left(s(A) \geq t\right) \geq 1 - \epsilon$.*

*Proof.* Our goal is to show that

$$P\left(s = \left[1 - \exp(-\lambda m)\right] \cdot \tfrac{A}{m} \geq t\right) \geq 1 - \epsilon, \tag{10}$$

where $A \sim \text{Binomial}(m, p)$ and $m$ is the number of (independent) subgraphs or subsamples considered. Rewriting $s \geq t$ gives

$$\left[1 - \exp(-\lambda m)\right] \cdot \frac{A}{m} \geq t \quad \Longleftrightarrow \quad \frac{A}{m} \geq \frac{t}{1 - \exp(-\lambda m)}.$$

For notational simplicity, we define $r = \frac{t}{1-\exp(-\lambda m)}$. Hence, our goal becomes ensuring $P(A \geq mr) \geq 1 - \epsilon$. Since $A$ is a binomial random variable $A \sim \text{Binomial}(m, p)$, $\mathbb{E}[A] = mp$, we therefore have the Chernoff bound, states that,

$$P\left(A \leq (1 - \delta)mp\right) \leq \exp\left(-\tfrac{\delta^2}{2}mp\right), \tag{11}$$

for any $0 < \delta < 1$. Subsequently, we set $(1 - \delta)mp = mr$, i.e., $\delta = 1 - \frac{r}{p}$. Note that for this $\delta$ to be positive (so that the Chernoff bound form applies), we need $r < p$. In other words,

$$\frac{t}{1 - \exp(-\lambda m)} = r < p,$$

which is the intuitive condition that the true probability $p$ exceeds the effective threshold $r$. With the above definition of $\delta$,

$$P\left(A < mr\right) = P\left(A \leq (1 - \delta)mp\right) \leq \exp\left(-\tfrac{mp}{2}\left(1 - \tfrac{r}{p}\right)^2\right).$$

Hence

$$P\left(A \geq mr\right) \geq 1 - \exp\left(-\tfrac{mp}{2}\left(1 - \tfrac{r}{p}\right)^2\right). \tag{12}$$

To ensure this probability is at least $1 - \epsilon$, we impose

$$\exp\left(-\tfrac{mp}{2}\left(1 - \tfrac{r}{p}\right)^2\right) \leq \epsilon.$$

Since $r = \frac{t}{1-e^{-\lambda m}}$, this condition explicitly becomes

$$\frac{mp}{2}\left(1 - \frac{t}{p(1 - e^{-\lambda m})}\right)^2 \geq \log \frac{1}{\epsilon}. \tag{13}$$

Therefore, whenever (3) and $r < p$ is satisfied, we have

$$P\left(\left[1 - e^{-\lambda m}\right] \cdot \tfrac{A}{m} \geq t\right) = P\left(A \geq mr\right) \geq 1 - \epsilon. \tag{14}$$

Hence the theorem follows. □

## C.3 Proof of Corollory 3.3

**Corollory 3.3** (Upper bound of node in subgraphs) *Let $\lambda > 0$, $t \in (0, 1)$, and $\epsilon \in (0, 1)$ be fixed. For a candidate edge $(X, Y)$, denote by $m$ the number of local subgraphs whose Markov Blankets contain both endpoints. Under the setting of Theorem 3.2, the sufficient condition (3) can be converted into the explicit bound*

$$m \geq \frac{2 \log(1/\epsilon)}{p\left((1 - t/p)^2 - 2(t/p)(1 - t/p)e^{-\lambda}\right)},$$

*Proof.* We first define $y = \exp(-\lambda m)$. Then, by the conclusion of Theorem 4.4, we obtain

$$-\frac{p}{2\lambda}\log y\left(1 - \frac{t}{p(1-y)}\right)^2 \geq \log\frac{1}{\epsilon}. \tag{15}$$

Next, we consider the first-order Taylor expansion:

$$\begin{aligned}
\left(1 - \frac{t}{p}\frac{1}{1-y}\right)^2 &= \left[1 - \frac{t}{p} - \frac{t}{p}y + O(y^2)\right]^2 \\
&= \left[\gamma - \theta y + O(y^2)\right]^2 \\
&= \gamma^2 - 2\theta\gamma y + O(y^2),
\end{aligned} \tag{16}$$

where we set $\theta = \frac{t}{p}$ and $\gamma = 1 - \frac{t}{p}$. Therefore, (15) becomes

$$\log y\left[\gamma^2 - 2\theta\gamma y + O(y^2)\right] \leq -\frac{2\lambda}{p}\log\frac{1}{\epsilon}. \tag{17}$$

Therefore, by substituting $\log y = -\lambda m$ and dropping the $O(y^2)$ term (since $y^2$ is small enough, generally less than $10^{-4}$), we get an approximate condition:

$$m(\gamma^2 - 2\theta\gamma e^{-\lambda m}) \geq \frac{2}{p}\log\frac{1}{\epsilon}. \tag{18}$$

This is an implicit condition on $m$. To derive an explicit and sufficient lower bound, we strengthen the left-hand side. Since $m \geq 1$, we have $e^{-\lambda m} \leq e^{-\lambda}$, therefore

$$\gamma^2 - 2\theta\gamma e^{-\lambda m} \geq \gamma^2 - 2\theta\gamma e^{-\lambda}.$$

Let $K_\lambda = \gamma^2 - 2\theta\gamma e^{-\lambda}$. To ensure the lower bound is positive, we require $\gamma^2 > 2\theta\gamma e^{-\lambda}$, or equivalently $\gamma > 2\theta e^{-\lambda}$ (since $\gamma = 1 - t/p > 0$). This condition simplifies to $1 - t/p > 2(t/p)e^{-\lambda}$.

By our Theorem 3.4, since $\lambda > \frac{1}{m}\log(1-t)$, we have

$$\frac{t}{p}\left(2e^{-\lambda} + 1\right) < \frac{t}{p}\left(2(1-t)^{\frac{1}{m}} + 1\right) < 1.$$

This inequality is easily satisfied and thus represents a very mild condition. Consequently, we can regard the lower bound $K_\lambda > 0$ as being established. Then the inequality (18) is satisfied under a stronger condition:

$$m \cdot K_\lambda \geq \frac{2}{p}\log\frac{1}{\epsilon}.$$

Solving for $m$ gives an explicit lower bound:

$$m \geq \frac{2\log(1/\epsilon)}{pK_\lambda} = \frac{2\log(1/\epsilon)}{p(\gamma^2 - 2\theta\gamma e^{-\lambda})}.$$

Substituting the definitions of $\gamma$ and $\theta$, we obtain:

$$m \geq \frac{2\log(1/\epsilon)}{p\left((1-t/p)^2 - 2(t/p)(1-t/p)e^{-\lambda}\right)}.$$

$\square$

# D   Discussion of the Structure-Aware Error Bound

The weighted voting procedure serves as the core mechanism for aggregating local subgraph estimates into a global DAG. While this method adjusts edge confidence based on empirical support, its effectiveness ultimately depends on the ability to balance false positives and false negatives across the merged graph. To better understand this behavior, we analyze the global error induced by the weighted voting rule and how it interacts with the sparsity of the graph, the choice of voting threshold, and the distribution of subgraph overlaps.

This section formalizes that analysis. We first derive a decomposition of the total error into false positive and false negative components, followed by a structure-aware upper bound based on the union bound. The role of the weighting parameter $\lambda$ is then examined in detail, culminating in formal proofs of Theorem 3.4 and Theorem 3.5, which establish a feasible range for $\lambda$ and the asymptotic vanishing of global error, respectively. These bounds are further instantiated under Erdős–Rényi (ER) and scale-free (SF) graph models to characterize how graph topology influences the merging accuracy.

To begin with, we formalize the decomposition of the global error into *false negatives* (FN) and *false positives* (FP), and derive a structure-aware upper bound based on the union bound. We summarized it into the following lemma:

**Lemma D.1** (Structure-aware global error bound). *Each candidate directed edge $(V_i, V_j)$ is evaluated in $m_{ij}$ independent local sub-graphs whose Markov Blankets contain both endpoints.*

- *For a **true** edge, the vote count obeys $A_{ij} \sim \mathrm{Binomial}(m_{ij}, p)$.*

- *For a **false** edge, $A_{ij} \sim \mathrm{Binomial}(m_{ij}, q)$ with $p > q$.*

*Using the weighted rule*

$$s_{ij} = \left[1 - e^{-\lambda m_{ij}}\right] \frac{A_{ij}}{m_{ij}} \geq t, \qquad r_\lambda(m_{ij}) = \frac{t}{1 - e^{-\lambda m_{ij}}},$$

*assume $p > r_\lambda(m_{ij})$ and $q < r_\lambda(m_{ij})$ for every edge. Then*

$$\Pr(\textit{global error}) \leq \underbrace{\sum_{(i,j) \in E^*} e^{-2m_{ij}[p - r_\lambda(m_{ij})]^2}}_{\textit{FN contribution}} + \underbrace{\sum_{(i,j) \notin E^*} e^{-2m_{ij}[r_\lambda(m_{ij}) - q]^2}}_{\textit{FP contribution}}, \tag{19}$$

*where $E^*$ denotes the ground-truth edge set.*

*Proof.* For $(V_i, V_j) \in \boldsymbol{E}^*$, we have

$$\Pr\big(\text{FN on } (V_i, V_j)\big) = \Pr\big(A_{ij}/m_{ij} < r_\lambda(m_{ij})\big) \leq e^{-2m_{ij}[p - r_\lambda(m_{ij})]^2} \tag{20}$$

by Hoeffding's inequality. A symmetric argument gives the FP term for $(V_i, V_j) \notin \boldsymbol{E}^*$. Finally, the union bound over all edges yields the claimed inequality. $\qquad\square$

**Corollary D.2** (Worst-case simplification). *If $m_{ij} \geq m_{\min}$ for all edges, then*

$$\Pr(\textit{global error}) \leq N_{FN}\, e^{-2m_{\min}\left[p - r_\lambda(m_{\min})\right]^2} + N_{FP}\, e^{-2m_{\min}\left[r_\lambda(m_{\min}) - q\right]^2}, \tag{21}$$

*where $N_{FN} = |\boldsymbol{E}^*|$ and $N_{FP} = \binom{n}{2} - N_{FN}$ for a graph with $n$ nodes.*

The error bound derived above depends on the effective threshold $r_\lambda(m)$, which is controlled by the weighting parameter $\lambda$. To understand the role of this parameter, it is instructive to consider the limiting case $\lambda = 0$, which corresponds to the naive voting scheme. In this case, the weight term disappears, and the edge inclusion rule reduces to comparing the raw directional frequency $A/m$ against the fixed threshold $t$.

*Remark* D.3 (Naive voting baseline). If we drop the weight and decide solely on the unweighted fraction $\frac{A_{ij}}{m_{ij}} \geq t$, Lemma D.1 specialises to

$$\Pr(\text{global error}) \leq \sum_{(i,j)\in E^*} e^{-2m_{ij}(p-t)^2} + \sum_{(i,j)\notin E^*} e^{-2m_{ij}(t-q)^2}. \tag{22}$$

In sparse graphs, where the number of candidate false positive edges vastly exceeds the number of true positives (i.e., $N_{\text{FP}} \gg N_{\text{FN}}$), the overall error is typically dominated by the first summation term. Therefore, a moderate increase in $\lambda$ can lead to a significant reduction in total error by aggressively penalizing low-support spurious edges, even if it slightly increases the false negative rate. This trade-off is particularly favorable in high-dimensional settings, where controlling the false discovery rate is often more critical than maximizing recall. These insights align with the empirical results reported in Section 4.1, where the weighted voting scheme consistently improves FDR without severely compromising TPR across a wide range of base learners.

## D.1 Influence and Practical Range of the Weight Parameter $\lambda$

To ensure that the weighted voting mechanism achieves a reliable trade-off between false positives and false negatives, it is necessary to understand how the choice of the weighting parameter $\lambda$ affects the acceptance threshold and the overall error bound. The following derivation provides a characterization of the feasible range of $\lambda$ that satisfies the conditions used in the theoretical analysis of edge decisions. This directly supports the proof of Theorem 3.4 in the main text.

**Theorem 3.4** (Practical choice of $\lambda$) *Fix a vote count $m \geq 1$, a decision threshold $t \in (0,1)$, and a target error level $\epsilon \in (0,1)$. If $\lambda$ satisfies*

$$-\frac{1}{m}\ln(1-t) < \lambda \leq -\frac{1}{m}\ln\epsilon,$$

*then the weighted-vote rule achieves the prescribed error control under the union bound.*

*Proof.* Define the Hoeffding-based global error upper bound $\mathcal{L}(\lambda) = N_{\text{FN}}\, e^{-2m_{\min}(p-r_\lambda)^2} + N_{\text{FP}}\, e^{-2m_{\min}(r_\lambda-q)^2}$, where $N_{\text{FN}}$ ($N_{\text{FP}}$) is the number of true (false) candidate edges rescaled by their respective cost coefficients. For notational simplicity, we omit the subscripts, and use $m$ to represent $m_{\min}$ in our later proof. We first differentiate $\mathcal{L}$ w.r.t. $\lambda$:

$$\begin{aligned} \frac{\partial r_\lambda}{\partial \lambda} &= \frac{tme^{-\lambda m}}{(1-e^{-\lambda m})^2} = r_\lambda \frac{me^{-\lambda m}}{1-e^{-\lambda m}} > 0, \\ \frac{\partial \mathcal{L}}{\partial \lambda} &= 2m\frac{te^{-\lambda m}}{(1-e^{-\lambda m})^2}\Big[N_{\text{FN}}\,\delta_p e^{-2m\delta_p^2} - N_{\text{FP}}\,\delta_q e^{-2m\delta_q^2}\Big], \end{aligned} \tag{23}$$

with $\delta_p = p - r_\lambda > 0$ and $\delta_q = r_\lambda - q > 0$.

Because $\delta_p$ *increases* and $\delta_q$ *decreases* as $\lambda$ grows, a larger $\lambda$ lowers the false-negative term (higher **recall**) but raises the false-positive term (lower **precision**). For sparse causal graphs we typically have $N_{\text{FP}} \gg N_{\text{FN}}$, making the second term dominant and hence $\partial\mathcal{L}/\partial\lambda < 0$ *until* the exponential weight saturates. Consequently, increasing $\lambda$ is beneficial *only* inside a finite interval.

**Upper bound for $\lambda$.** In the worst-case scenario where all candidate edges are consistently supported in the same direction, the voting scores for both true and false edges become uniformly close to $1 - e^{-\lambda m}$. If $1 - e^{-\lambda m} \geq 1 - \epsilon$ ($0 < \epsilon \ll 1$), then even false edges can exceed the decision threshold $t$, leading to a large number of number of false positives. Therefore To avoid such indiscriminate acceptance, $\lambda$ must be chosen to ensure that $1 - e^{-\lambda m}$ remains sufficiently below 1. Solving $e^{-\lambda m} = \epsilon$ gives

$$0 < \lambda \leq \lambda_{\max}(\epsilon) = -\frac{1}{m}\ln\epsilon. \tag{24}$$

**Lower bound for $\lambda$.** The effective threshold $r_\lambda(m) = \dfrac{t}{1 - e^{-\lambda m}}$ must satisfy $0 < r_\lambda < 1$; otherwise the acceptance condition $A/m \geq r_\lambda$ can never be met because $A/m \leq 1$ by definition. Solving the inequality $r_\lambda < 1$ yields

$$\lambda > \lambda_{\min} := -\frac{1}{m} \ln(1 - t). \tag{25}$$

Intuitively, when $\lambda$ falls below this bound the exponential weight is so close to 1 that the prefactor $1 - e^{-\lambda m}$ becomes *smaller* than $t$, inflating $r_\lambda$ beyond 1 and blocking every candidate edge, including true ones. Hence $\lambda_{\min}$ is the *viability threshold*: only for $\lambda > \lambda_{\min}$ does the weighted-voting rule retain a non-zero recall. Therefore a practical search range is

$$\lambda \in \left[ -\frac{1}{m} \ln(1 - t), \ \frac{1}{m} \ln \epsilon \right], \tag{26}$$

within which cross-validation or the closed-form condition $\partial \mathcal{L} / \partial \lambda = 0$ can be used to pinpoint an optimal $\lambda^\star$. □

**Exponentially vanishing reversal error.** For any $\lambda$ in this range and any true edge with support probability $p > r_\lambda$, the probability of being accepted in the reverse direction is $\Pr(\text{reverse}) \leq \exp[-2m \, (p - r_\lambda)^2]$, which decays exponentially with the number of independent subgraphs $m$. This guarantees that the weighted-voting merger remains statistically consistent as data grow, while a properly chosen $\lambda$ suppresses spurious edges in finite-sample regimes.

The general error bound depends on the number of subgraphs in which each edge appears. This quantity is influenced by the underlying graph topology. In the following, the behavior of the bound is examined under two commonly used random graph models: Erdős–Rényi and scale-free graphs. The analysis characterizes typical support counts and their implications for the error terms derived in Lemma D.1.

## D.2 Erdős–Rényi and Scale-Free Graphs

The error bounds derived in the previous section depend not only on the weighting parameter $\lambda$, but also on the empirical support count $m_{ij}$ which measures the number of subgraphs in which each edge appear. This quantity is influenced by the underlying graph topology and the statistical properties of the Markov Blanket construction.

To understand how $m_{ij}$ behaves in practice, we analyze two representative random graph models: Erdős–Rényi (ER) and scale-free (SF) networks. These models differ significantly in their degree distributions, which in turn affect the overlap patterns among Markov Blankets and the expected frequency with which edges are covered by local subgraphs. The analysis below characterizes typical support rates under each model, providing context for interpreting the global error bounds and informing the expected sample complexity of reliable aggregation.

**Theorem D.4** (ER-$h$ graph)**.** *Let $G \sim ER(n, \theta)$ with edge probability $\theta = h/(n-1)$, and assign directions by a random topological order so that the expected out-degree is $h$. Denote*

$$\delta_p := p - r_\lambda(2), \qquad \delta_q := r_\lambda(2) - q \quad (\delta_p, \delta_q > 0).$$

*Then, with probability at least $1 - O(\theta^2)$ over the graph draw,*

$$\Pr(\text{global error}) \leq \frac{nh}{2} \, e^{-4\delta_p^2} + \frac{n(n-1) - nh}{2} \, e^{-4\delta_q^2} + O(\theta^2). \tag{27}$$

*The $O(\theta^2)$ term covers the negligible fraction of edges whose vote count $m_{ij} > 2$.*

*Proof.* In a directed ER graph each vertex has $\deg^{\text{in}}, \deg^{\text{out}} \sim \text{Pois}(\theta/2)$, so $\mathbb{E}\Big[|\text{MB}(v)|\Big] = \mathbb{E}[\deg^{\text{in}} + \deg^{\text{out}} + \text{spouses}] \approx 2h$, where the "spouse" term (*co-parents*) shares the same mean as $\deg^{\text{out}}$.

For an oriented edge $(i, j)$, it appears in *both* $\mathrm{MB}(V_i)$ and $\mathrm{MB}(V_j)$, giving a baseline $m_{ij} \geq 2$. Additionally, it appears in $\mathrm{MB}(V_k)$ for every common child $V_k$ of $V_i$ and $V_j$. For fixed $V_k$, the events "$V_i \to V_k$" and "$V_j \to V_k$" are independent with probability $\theta^2$. Hence the number of common children follows $\mathrm{Pois}(\lambda_c)$ with $\lambda_c = (n-2)\theta^2 \approx h^2/n$.

Thus,

$$m_{ij} = 2 + X, \qquad X \sim \mathrm{Pois}(\lambda_c).$$

When $h = O(1)$, $\lambda_c = O(\theta^2) \ll 1$, whence

$$\Pr(m_{ij} = 2) = 1 - O(\theta^2), \qquad \Pr(m_{ij} \geq 3) = O(\theta^2).$$

For the overwhelming majority of edges ($m_{ij} = 2$), lemma D.1 gives:

$$\Pr(\text{FN on } (i,j)) \leq e^{-4\delta_p^2}, \qquad \Pr(\text{FP on } (i,j)) \leq e^{-4\delta_q^2}.$$

Counting edges:

$$N_{\mathrm{FN}} \approx \frac{nh}{2}, \qquad N_{\mathrm{FP}} = \binom{n}{2} - N_{\mathrm{FN}}.$$

Summing the two contributions yields

$$\frac{nh}{2} e^{-4\delta_p^2} + \frac{n(n-1) - nh}{2} e^{-4\delta_q^2}. \tag{28}$$

$\square$

We can obtain similar results from the SF graph.

**Theorem D.5** (SF-$h$ graph). *Let $\mathcal{G}$ be a directed* scale-free *graph on $n$ nodes, obtained by sampling an undirected Chung–Lu (or Barabási–Albert) graph whose degree sequence $(d_1, \dots, d_n)$ satisfies*

$$\Pr(d \geq k) \leq C_\alpha k^{1-\alpha}, \qquad 2 < \alpha < 3, \tag{29}$$

*and whose* mean degree *is $h$; and orienting edges according to a random topological order. Then, for a universal constant $C_\alpha$ that depends only on $\alpha$,*

$$\Pr(\text{global error}) \leq \frac{nh}{2} e^{-4\delta_p^2} + \frac{n(n-1)}{2} - \frac{nh}{2} e^{-4\delta_q^2} + \frac{n(n-1)}{2} C_\alpha n^{-(\alpha-2)}. \tag{30}$$

*Proof.* For an oriented edge $(V_i, V_j)$ let $d_i, d_j$ be its endpoint degrees. Exactly as in the ER case each edge appears at least twice; additional occurrences come from every *common child* $k$ with probability $(d_i/n)(d_j/n)$. Hence

$$m_{ij} = 2 + X, \quad X \sim \mathrm{Pois}\left(\lambda_{ij}\right), \quad \lambda_{ij} := \frac{d_i d_j}{n}.$$

For any fixed $\lambda$ and $\delta \in \{\delta_p, \delta_q\}$

$$\begin{aligned}
\mathbb{E}\left[e^{-2m_{ij}\delta^2} \mid \lambda_{ij}\right] &= e^{-4\lambda_{ij}^2} \mathbb{E}\left[e^{-2X\delta^2}\right] \\
&= e^{-4\delta^2} e^{\lambda_{ij}(e^{-2\delta^2} - 1)}.
\end{aligned} \tag{31}$$

Since the value of $e^{-2\delta^2} - 1$ varies, we splitted the expectation (31) into two regimes:

$$\mathbb{E}\left[e^{-2m_{ij}\delta^2}\right] = \mathbb{E}\left[e^{-2m_{ij}\delta^2} \mathbf{1}_{\{\lambda_{ij} \leq 1\}}\right] + \mathbb{E}\left[e^{-2m_{ij}\delta^2} \mathbf{1}_{\{\lambda_{ij} > 1\}}\right]. \tag{32}$$

- Non-hub regime $\lambda_{ij} \leq 1$:

$$\mathbb{E}\big[e^{-2m_{ij}\delta^2} \mid \lambda_{ij}\big] \leq e^{-4\delta^2}. \tag{33}$$

- Hub regime $\lambda_{ij} > 1$. By (31) the conditional term is $\leq e^{-4\delta^2} e^{-\lambda_{ij}/2} \leq 1$, but the probability of this event can be bounded with the degree tail:

$$\mathbb{E}\big[e^{-2m_{ij}\delta^2} \mathbf{1}_{\{\lambda_{ij}>1\}}\big] \leq \Pr\big(\lambda_{ij} > 1\big) = \Pr\Big(\frac{d_i d_j}{n} > 1\Big) \leq C_\alpha n^{-(\alpha-2)},$$

  where we apply the union bound to decompose the event $d_i d_j > n$ into two simpler events, $d_i > n^{1/2}$ or $d_j > n^{1/2}$, control each using the degree tail bound $\Pr(d \geq k) \leq C_\alpha k^{1-\alpha}$, and then combine the two estimates.

Therefore, for either $\delta = \delta_p$ or $\delta_q$,

$$\mathbb{E}\big[e^{-2m_{ij}\delta^2}\big] \leq e^{-4\delta^2} + C_\alpha n^{-(\alpha-2)}. \tag{34}$$

There are $N_{\mathrm{FN}} \approx nh/2$ true and $N_{\mathrm{FP}} = \binom{n}{2} - N_{\mathrm{FN}}$ false edges on average. Multiplying the expectation (34) by these counts and plugging into Lemma D.1 yields inequality (30). □

Theorem D.5 completes the structure-aware error analysis by characterizing the influence of heterogeneous degree distributions on the residual error bound. While the dominant exponential terms governing false positive and false negative rates are structurally similar to those in Theorem D.4, the residual term exhibits a slower decay due to the presence of high-degree nodes. These hub-related structures lead to greater variability in the support count $m_{ij}$ across candidate edges.

This variability has practical implications. In networks where edge supports are highly non-uniform, the weighted voting mechanism implicitly induces a form of confidence calibration: high-support edges, typically associated with structurally central nodes, retain larger weights and are more likely to be preserved. In contrast, low-support edges often arising from sparse or weakly connected regions, will be heavily penalized by the exponential weighting term. This differential treatment improves robustness to statistical noise and helps suppress false positives without uniformly raising the threshold for all decisions.

As a result, the error reduction effect of the weighting scheme is not solely determined by the average support level, but also by the variance in subgraph overlap. Networks with broader support distributions provide more opportunities for selective edge retention, which enhances the overall effectiveness of the aggregation procedure. This observation complements the earlier asymptotic result, and offers a finer-grained explanation of the empirical precision gains observed in our experiments.

To complete the analysis, we examine how the global error behaves asymptotically under increasing graph size.

### D.3 Asymptotic Analysis

**Theorem 3.5** (Asymptotic Consistency) *Fix a threshold $t \in (0,1)$ and let $\delta_p = p - t$ and $\delta_q = t - q$ denote the positive margins between $t$ and the inclusion probabilities $p, q$ of true and false edges respectively. Assume $\delta_p, \delta_q > 0$ and that $\lambda$ satisfies the conditions in Theorem 3.4. If every candidate pair receives $m_{ij} \geq m = C \log n$ directional votes with $C > \max\{\frac{1}{2\delta_p^2}, \frac{1}{\delta_q^2}\}$, then we have*

$$\Pr(\text{global error}) = o(1), \qquad \text{as } n \to \infty. \tag{35}$$

*Proof.* By the conclusion of Lemma D.1, the global error probability is bounded by

$$\Pr(\text{global error}) \leq \sum_{(i,j)\in E^*} e^{-2m_{ij}(p-t)^2} + \sum_{(i,j)\notin E^*} e^{-2m_{ij}(t-q)^2}. \tag{36}$$

Since the number of true edges satisfies $N_{\text{FN}} = |\boldsymbol{E}^*| = \mathcal{O}(n)$, and the number of false edges is $N_{\text{FP}} = \binom{n}{2} - N_{\text{FN}} = \mathcal{O}(n^2)$, we can simplify the above bound by letting $m_{ij} \equiv m$ for all edges:

$$\Pr(\text{global error}) \leq N_{\text{FN}} \, e^{-2m\delta_p^2} + N_{\text{FP}} \, e^{-2m\delta_q^2},$$

where we denote $\delta_p = p - t > 0$ and $\delta_q = t - q > 0$.

Note that the margins in Lemma D.1 are measured with respect to $r_\lambda(m)$ rather than $t$; since $r_\lambda(m) > t$, the false-negative margin $p - r_\lambda(m)$ is *smaller* than $\delta_p$ and cannot simply be replaced by it, so we control the gap explicitly.

*False positives.* Since $r_\lambda(m) \geq t$ we have $r_\lambda(m) - q \geq \delta_q$ (in particular $q < r_\lambda(m)$, so the false-edge condition of Lemma D.1 holds), and therefore

$$N_{\text{FP}} \, e^{-2m[r_\lambda(m)-q]^2} \ \leq \ \mathcal{O}(n^2) \cdot n^{-2C\delta_q^2} \ = \ \mathcal{O}\!\left(n^{2-2C\delta_q^2}\right) \ \longrightarrow \ 0,$$

because $C > 1/\delta_q^2$ is equivalent to $2 - 2C\delta_q^2 < 0$.

*False negatives.* For $n$ large enough that $e^{-\lambda m} \leq \frac{1}{2}$,

$$0 \ \leq \ r_\lambda(m) - t \ = \ \frac{t \, e^{-\lambda m}}{1 - e^{-\lambda m}} \ \leq \ 2t \, e^{-\lambda m} \ = \ 2t \, n^{-\lambda C} \ \longrightarrow \ 0.$$

Because the condition $C > \frac{1}{2\delta_p^2}$ is strict, we may fix $\zeta \in (0, \delta_p)$ with $C > \frac{1}{2(\delta_p - \zeta)^2}$. For all $n$ large enough that $2t \, n^{-\lambda C} \leq \zeta$, we have $r_\lambda(m) \leq t + \zeta < p$ — so the true-edge condition $p > r_\lambda(m)$ of Lemma D.1 holds, as does the admissibility requirement $r_\lambda(m) < 1$ — and consequently $p - r_\lambda(m) \geq \delta_p - \zeta > 0$. Hence

$$N_{\text{FN}} \, e^{-2m[p-r_\lambda(m)]^2} \ \leq \ \mathcal{O}(n) \cdot n^{-2C(\delta_p-\zeta)^2} \ = \ \mathcal{O}\!\left(n^{1-2C(\delta_p-\zeta)^2}\right) \ \longrightarrow \ 0.$$

Combining the two displays completes the proof. $\qquad\square$

**Complexity.**   Finally, we analyze the computational complexity, which consists of two parts:

- The local structure learning phase takes $\mathcal{O}(m^3)$ per node, and there are $n$ nodes, resulting in $\mathcal{O}(nm^3)$ total cost.

- The voting and merging phase requires computing pairwise edge counts and resolving cycles over $\mathcal{O}(n^2)$ edge pairs, leading to an additional $\mathcal{O}(n^2)$ term.

Substituting $m = \mathcal{O}(\log n)$, the total runtime becomes

$$\mathcal{O}\left(n(\log n)^3 + n^2\right) = \tilde{\mathcal{O}}(n^2),$$

where the soft-$\mathcal{O}$ notation hides polylogarithmic factors. Thus, the proposed divide-and-conquer method achieves both statistical consistency and near-quadratic scalability.

# E   Implementation Details

Our code is based on two open-source packages: `gcastle`, which provides implementations of score-based and continuous causal discovery methods such as NOTEARS, GOLEM, GraN-DAG and DAG-GNN, and `dodiscover`, which implements ordering-based methods. These packages form the backbone of our experimental framework. On top of them, we implement our own modules for subgraph construction, weighted voting aggregation, and cycle removal. The full pipeline with configuration scripts and reproducibility controls is described in detail below. Subsequent subsections provide additional implementation details for baseline configuration, extended experimental results, runtime breakdown, and comparison against DCILP.

### E.1 Baselines

All baseline methods are implemented using publicly available code and configured with recommended hyperparameters. For methods involving continuous optimization, the primary computational bottleneck lies in gradient-based acyclicity constraints, which require $\mathcal{O}(d^3)$ time and $\mathcal{O}(d^2)$ memory due to matrix operations over the full graph. Discrete search–based methods such as SCORE and CAM incur combinatorial overhead when handling larger node counts. In all cases, integrating these methods into the VISTA framework significantly reduces both runtime and memory usage, as the local subgraphs are orders of magnitude smaller and can be processed independently.

**NOTEARS**  This method reformulates the combinatorial problem of DAG structure learning into a purely continuous optimization problem. It introduces a novel, smooth, and exact characterization of acyclicity using a matrix exponential function $h(W) = tr(W \circ W) - d = 0$. This transformation allows the problem to be solved efficiently using standard gradient-based optimization techniques, avoiding discrete search over graph structures.

**GOLEM**  This work analyzes the role of sparsity and DAG constraints in learning linear DAGs, noting potential optimization issues with hard DAG constraints required by prior methods like NOTEARS. It proposes GOLEM (Gradient-based Optimization of dag-penalized Likelihood for learning linEar dag Models), which uses a likelihood-based score function instead of least squares. The key finding is that applying soft sparsity and DAG penalties to this likelihood objective suffices to recover the ground truth DAG structure asymptotically, resulting in an unconstrained optimization problem that is easier to solve.

**DAG-GNN**  This method employs a deep generative model, specifically a Variational Autoencoder (VAE), to learn DAG structures, extending beyond linear models. It parameterizes the VAE's encoder and decoder using a novel Graph Neural Network (GNN) architecture, designed to capture complex non-linear relationships inherent in data. The approach learns the graph's weighted adjacency matrix alongside the neural network parameters, enforcing acyclicity through a continuous polynomial constraint, and naturally handles both continuous and discrete variables.

**GraN-DAG**  This work extends continuous DAG learning to nonlinear settings by parameterizing each conditional distribution with neural networks and constructing a weighted adjacency matrix from network connectivity. Acyclicity is enforced through a smooth matrix-exponential constraint, enabling gradient-based optimization of the likelihood objective. Post-processing with thresholding and pruning helps recover sparse graphs.

**SCORE**  This method recovers causal graphs for non-linear additive noise models by utilizing the score function $(\nabla \log p(x))$ of the observational data distribution. It establishes that the Jacobian of the score function reveals information sufficient to identify leaf nodes in the causal graph. By iteratively identifying and removing leaves based on the variance of the score's Jacobian diagonal elements, a topological ordering is estimated. The SCORE algorithm employs score matching techniques, specifically an extension of Stein's identity, to approximate the necessary score Jacobian components from data samples.

**CAM**  This approach estimates additive SEMs by decoupling the task into order search and edge selection. It first estimates a causal ordering of the variables using (potentially restricted) maximum likelihood, exploiting the identifiability property of additive models. Given the estimated order, sparse additive regression methods are then applied to select relevant parent variables (edges) for each node and estimate the corresponding additive functions. For high-dimensional data, an initial neighborhood selection step can reduce the search space before estimating the order.

In addition to the above baselines, we also include DCILP Dong et al. (2024), a recently proposed divide-and-conquer method that combines Markov Blanket estimation with global structure recovery via integer linear programming. While DCILP shares a similar high-level motivation with VISTA, it suffers from several practical limitations. Most notably, its final merging step relies on solving a large-scale ILP problem, which becomes computationally infeasible as the graph size increases. In many of our experimental settings, DCILP

either fails to complete within a reasonable time window or produces no feasible solution at all. These issues highlight the need for a more lightweight and scalable integration procedure, which motivates the design of VISTA. We provide a direct comparison with DCILP in the following section.

## E.2  Time comparison

Since each local structure is learned independently based on a variable's Markov Blanket, the entire divide phase can be executed in parallel across variables or computing nodes. This distributed strategy significantly reduces total runtime, especially when base learners are computationally intensive, such as neural network based models such as DAG-GNN and GraN-DAG or algorithms involving topological sorting such as SCORE.

Table 7, 8 and 9 confirm that VISTA consistently reduces the total execution time across a variety of settings. In large-scale graphs, where direct application of base methods may be computationally prohibitive, our framework provides a scalable alternative that decomposes the original problem into tractable subproblems. The integration step is lightweight and adds negligible overhead relative to the base learners. These results demonstrate that the benefits of VISTA are not limited to statistical performance, but also extend to practical runtime efficiency, enabling the application of complex causal discovery methods to larger and more realistic graphs.

Table 7: Comparison of total computing time (s) under ER5 setting.

| Method | $n = 50$ | $n = 100$ | $n = 300$ |
|---|---|---|---|
| NOTEARS | $510.73 \pm 84.15$ | $2465.33 \pm 58.02$ | $22407.77 \pm 940.32$ |
| +VISTA | $\mathbf{213.73 \pm 149.68}$ | $\mathbf{1096.51 \pm 142.87}$ | $\mathbf{3714.30 \pm 908.81}$ |
| GOLEM | $76.16 \pm 7.59$ | $115.01 \pm 35.82$ | $276.80 \pm 11.03$ |
| +VISTA | $\mathbf{23.25 \pm 0.67}$ | $\mathbf{37.57 \pm 1.36}$ | $\mathbf{46.53 \pm 4.61}$ |
| DAG-GNN | $794.42 \pm 72.61$ | $3137.68 \pm 214.75$ | $29801.46 \pm 1105.64$ |
| +VISTA | $\mathbf{311.34 \pm 54.23}$ | $\mathbf{818.52 \pm 501.88}$ | $\mathbf{3313.86 \pm 945.29}$ |
| GraN-DAG | $919.26 \pm 106.65$ | $5613.13 \pm 1068.14$ | $25684.95 \pm 2035.14$ |
| +VISTA | $\mathbf{208.43 \pm 26.62}$ | $\mathbf{934.72 \pm 50.64}$ | $\mathbf{2851.04 \pm 376.84}$ |
| SCORE | $629.88 \pm 93.72$ | $15876.42 \pm 807.89$ | ——— |
| +VISTA | $\mathbf{191.84 \pm 33.49}$ | $\mathbf{479.60 \pm 38.19}$ | $\mathbf{945.45 \pm 72.27}$ |

Table 8: Comparison of total computing time (s) under SF3 setting.

| Method | $n = 50$ | $n = 100$ | $n = 300$ |
|---|---|---|---|
| NOTEARS | $713.30 \pm 58.81$ | $2813.36 \pm 804.27$ | $16631.62 \pm 632.76$ |
| +VISTA | $\mathbf{400.82 \pm 83.65}$ | $\mathbf{652.59 \pm 57.99}$ | $\mathbf{1714.09 \pm 237.32}$ |
| GOLEM | $100.73 \pm 45.25$ | $169.20 \pm 16.68$ | $398.58 \pm 45.03$ |
| +VISTA | $\mathbf{23.63 \pm 1.61}$ | $\mathbf{35.47 \pm 2.26}$ | $\mathbf{60.20 \pm 13.66}$ |
| DAG-GNN | $697.78 \pm 93.37$ | $3555.95 \pm 2050.91$ | $21242.03 \pm 2178.95$ |
| +VISTA | $\mathbf{282.70 \pm 297.75}$ | $\mathbf{645.77 \pm 308.88}$ | $\mathbf{2020.79 \pm 811.42}$ |
| GraN-DAG | $890.96 \pm 135.84$ | $4978.95 \pm 656.25$ | $19372.84 \pm 3037.94$ |
| +VISTA | $\mathbf{319.29 \pm 389.88}$ | $\mathbf{817.63 \pm 145.63}$ | $\mathbf{2849.62 \pm 1558.40}$ |
| SCORE | $495.12 \pm 62.44$ | $18643.16 \pm 970.22$ | ——— |
| +VISTA | $\mathbf{153.65 \pm 46.35}$ | $\mathbf{354.37 \pm 86.45}$ | $\mathbf{5080.02 \pm 3674.36}$ |

Table 9: Comparison of total computing time (s) under SF5 setting.

| Method | $n = 50$ | $n = 100$ | $n = 300$ |
|---|---|---|---|
| NOTEARS | $808.15 \pm 102.23$ | $2842.83 \pm 312.22$ | $18676.80 \pm 6873.83$ |
| +VISTA | $\mathbf{501.96 \pm 62.14}$ | $\mathbf{1200.41 \pm 536.14}$ | $\mathbf{3041.62 \pm 1003.68}$ |
| GOLEM | $77.82 \pm 11.90$ | $217.95 \pm 73.36$ | $446.16 \pm 30.04$ |
| +VISTA | $\mathbf{23.71 \pm 2.37}$ | $\mathbf{99.20 \pm 132.68}$ | $\mathbf{167.89 \pm 214.19}$ |
| DAG-GNN | $911.14 \pm 315.63$ | $5762.00 \pm 1714.01$ | $31106.3 \pm 452.12$ |
| +VISTA | $\mathbf{356.46 \pm 101.18}$ | $\mathbf{1133.18 \pm 306.12}$ | $\mathbf{2641.62 \pm 541.84}$ |
| GraN-DAG | $853.75 \pm 98.54$ | $4944.93 \pm 2325.58$ | $38163.22 \pm 3919.71$ |
| +VISTA | $\mathbf{313.24 \pm 120.79}$ | $\mathbf{934.83 \pm 218.82}$ | $\mathbf{2999.39 \pm 485.66}$ |
| SCORE | $637.37 \pm 48.40$ | $18904.31 \pm 344.10$ | —— |
| +VISTA | $\mathbf{187.91 \pm 38.43}$ | $\mathbf{2003.45 \pm 882.48}$ | $\mathbf{4124.09 \pm 1311.74}$ |

### E.3 Additional experiments

To assess the effectiveness and scalability of VISTA, we conduct extensive experiments across a diverse set of synthetic graph families, varying both in size and structural complexity. This part is a detailed supplement of our Section 4.1. Specifically, we evaluate performance on 14 different graph configurations derived from ER and SF graphs, each instantiated with average degrees of 3 and 5, and node sizes $n \in \{30, 50, 100, 300\}$. This results in a comprehensive benchmark covering both sparse and dense regimes under varying dimensionalities. For each configuration, we benchmark recent representative causal discovery methods. Each method is tested under three settings: the original baseline, VISTA with naive voting (+VISTA-NV), and VISTA with weighted voting (+VISTA-WV). Notably, CAM does not scale well with graph size and GraN-DAG fails when $n$ reaches 300, so we do not report the results here. Due to the increasing computational cost with graph size, we ran each experimental configuration 10 times for $n = 30$ and $n = 50$, 5 times for $n = 100$, and 3 times for $n = 300$, and report the average and standard deviation across trials.

Although the advantages of VISTA are most pronounced in high-dimensional or structurally complex settings, it is important to note that for some small-scale graphs, particularly relative sparse configurations such as ER3 with low node counts, the original base learners already achieve high accuracy. In these cases, the benefits of decomposition are less clear. Errors introduced during Markov Blanket identification and aggregation, as analyzed in Appendix C, may offset any gains from the divide-and-conquer process. When the true structure is relatively simple and well-recovered by the base model, additional processing may be unnecessary.

By contrast, as the graph size increases, structural coverage from local subgraphs becomes more reliable, and the advantages of localized inference and confidence-aware aggregation become more pronounced. In particular, VISTA consistently improves structural accuracy and reduces false discoveries for base learners that face scalability challenges in large and complex graphs, providing a practical approach to mitigating the curse of dimensionality in causal structure learning.

The remaining experimental results are as follows:

Table 10: Results with linear and nonlinear synthetic datasets ($n = 30$, $h = 5$).

| Method | ER5 | | | | SF5 | | | |
|---|---|---|---|---|---|---|---|---|
| | FDR↓ | TPR↑ | SHD↓ | F1↑ | FDR↓ | TPR↑ | SHD↓ | F1↑ |
| NOTEARS | $0.21 \pm 0.08$ | $0.73 \pm 0.07$ | $64.70 \pm 18.22$ | $0.76 \pm 0.05$ | $0.24 \pm 0.13$ | $0.70 \pm 0.10$ | $64.30 \pm 29.24$ | $0.73 \pm 0.08$ |
| +**VISTA-NV** | $0.63 \pm 0.00$ | $\mathbf{0.97 \pm 0.01}$ | $236.60 \pm 7.32$ | $0.53 \pm 0.01$ | $0.56 \pm 0.02$ | $\mathbf{0.98 \pm 0.01}$ | $155.20 \pm 11.90$ | $0.61 \pm 0.02$ |
| +**VISTA-WV** | $\mathbf{0.12 \pm 0.04}$ | $0.75 \pm 0.03$ | $\mathbf{50.00 \pm 8.64}$ | $\mathbf{0.81 \pm 0.03}$ | $\mathbf{0.03 \pm 0.02}$ | $0.86 \pm 0.03$ | $\mathbf{20.20 \pm 2.62}$ | $\mathbf{0.84 \pm 0.02}$ |
| GOLEM | $0.24 \pm 0.08$ | $0.79 \pm 0.07$ | $63.50 \pm 23.61$ | $0.77 \pm 0.05$ | $0.15 \pm 0.11$ | $0.85 \pm 0.09$ | $35.30 \pm 23.39$ | $0.85 \pm 0.07$ |
| +**VISTA-NV** | $0.65 \pm 0.01$ | $\mathbf{0.97 \pm 0.01}$ | $251.00 \pm 2.62$ | $0.52 \pm 0.01$ | $0.56 \pm 0.02$ | $\mathbf{0.99 \pm 0.01}$ | $158.40 \pm 3.09$ | $0.61 \pm 0.02$ |
| +**VISTA-WV** | $\mathbf{0.17 \pm 0.03}$ | $0.79 \pm 0.05$ | $\mathbf{53.00 \pm 8.73}$ | $\mathbf{0.81 \pm 0.04}$ | $\mathbf{0.01 \pm 0.01}$ | $0.89 \pm 0.03$ | $\mathbf{15.00 \pm 4.32}$ | $\mathbf{0.94 \pm 0.02}$ |
| DAG-GNN | $0.29 \pm 0.06$ | $0.77 \pm 0.19$ | $77.50 \pm 20.08$ | $0.74 \pm 0.09$ | $0.29 \pm 0.19$ | $0.72 \pm 0.25$ | $65.50 \pm 36.60$ | $0.72 \pm 0.16$ |
| +**VISTA-NV** | $0.64 \pm 0.00$ | $\mathbf{0.98 \pm 0.01}$ | $250.30 \pm 4.78$ | $0.52 \pm 0.00$ | $0.60 \pm 0.02$ | $\mathbf{0.99 \pm 0.01}$ | $189.70 \pm 14.06$ | $0.56 \pm 0.02$ |
| +**VISTA-WV** | $\mathbf{0.27 \pm 0.03}$ | $0.84 \pm 0.07$ | $\mathbf{60.00 \pm 7.85}$ | $\mathbf{0.78 \pm 0.05}$ | $\mathbf{0.03 \pm 0.02}$ | $0.87 \pm 0.04$ | $\mathbf{20.00 \pm 7.26}$ | $\mathbf{0.92 \pm 0.03}$ |
| CAM | $0.77 \pm 0.04$ | $0.53 \pm 0.08$ | $267.50 \pm 23.32$ | $0.32 \pm 0.04$ | $0.77 \pm 0.06$ | $0.54 \pm 0.11$ | $241.20 \pm 37.51$ | $0.32 \pm 0.06$ |
| +**VISTA-NV** | $0.78 \pm 0.04$ | $\mathbf{0.66 \pm 0.12}$ | $327.00 \pm 24.39$ | $0.33 \pm 0.06$ | $0.82 \pm 0.02$ | $0.57 \pm 0.07$ | $335.60 \pm 9.06$ | $0.27 \pm 0.03$ |
| +**VISTA-WV** | $\mathbf{0.63 \pm 0.08}$ | $0.40 \pm 0.10$ | $\mathbf{158.00 \pm 16.87}$ | $\mathbf{0.39 \pm 0.09}$ | $\mathbf{0.17 \pm 0.04}$ | $0.59 \pm 0.05$ | $\mathbf{122.00 \pm 5.35}$ | $\mathbf{0.69 \pm 0.04}$ |
| GraN-DAG | $0.67 \pm 0.12$ | $0.18 \pm 0.10$ | $159.60 \pm 24.52$ | $0.22 \pm 0.10$ | $0.72 \pm 0.38$ | $0.26 \pm 0.03$ | $187.80 \pm 80.73$ | $0.27 \pm 0.18$ |
| +**VISTA-NV** | $0.90 \pm 0.08$ | $\mathbf{0.39 \pm 0.12}$ | $211.70 \pm 49.91$ | $0.16 \pm 0.10$ | $0.85 \pm 0.29$ | $\mathbf{0.41 \pm 0.15}$ | $239.50 \pm 46.61$ | $0.22 \pm 0.31$ |
| +**VISTA-WV** | $\mathbf{0.31 \pm 0.06}$ | $0.14 \pm 0.07$ | $\mathbf{134.50 \pm 35.55}$ | $\mathbf{0.23 \pm 0.09}$ | $\mathbf{0.25 \pm 0.10}$ | $0.18 \pm 0.05$ | $\mathbf{128.60 \pm 52.47}$ | $\mathbf{0.29 \pm 0.07}$ |
| SCORE | $0.66 \pm 0.08$ | $0.43 \pm 0.05$ | $117.40 \pm 47.71$ | $0.38 \pm 0.05$ | $0.55 \pm 0.40$ | $0.71 \pm 0.24$ | $153.30 \pm 76.60$ | $0.55 \pm 0.31$ |
| +**VISTA-NV** | $0.80 \pm 0.06$ | $\mathbf{0.83 \pm 0.05}$ | $399.60 \pm 36.65$ | $0.32 \pm 0.08$ | $0.76 \pm 0.25$ | $\mathbf{0.88 \pm 0.04}$ | $440.60 \pm 49.79$ | $0.38 \pm 0.31$ |
| +**VISTA-WV** | $\mathbf{0.34 \pm 0.09}$ | $0.56 \pm 0.08$ | $\mathbf{95.50 \pm 28.86}$ | $\mathbf{0.61 \pm 0.06}$ | $\mathbf{0.36 \pm 0.16}$ | $0.79 \pm 0.05$ | $\mathbf{88.80 \pm 11.60}$ | $\mathbf{0.71 \pm 0.10}$ |

Table 11: Results with linear and nonlinear synthetic datasets ($n = 50$, $h = 3$).

| Method | ER3 | | | | SF3 | | | |
|---|---|---|---|---|---|---|---|---|
| | FDR↓ | TPR↑ | SHD↓ | F1↑ | FDR↓ | TPR↑ | SHD↓ | F1↑ |
| NOTEARS | $\mathbf{0.09 \pm 0.08}$ | $0.90 \pm 0.05$ | $\mathbf{24.90 \pm 17.75}$ | $\mathbf{0.90 \pm 0.05}$ | $0.22 \pm 0.13$ | $0.76 \pm 0.11$ | $62.80 \pm 33.53$ | $0.77 \pm 0.08$ |
| +**VISTA-NV** | $0.72 \pm 0.03$ | $\mathbf{0.97 \pm 0.01}$ | $353.40 \pm 57.19$ | $0.44 \pm 0.04$ | $0.68 \pm 0.04$ | $\mathbf{0.97 \pm 0.01}$ | $304.40 \pm 52.48$ | $0.48 \pm 0.04$ |
| +**VISTA-WV** | $\mathbf{0.09 \pm 0.03}$ | $0.86 \pm 0.03$ | $30.50 \pm 5.43$ | $0.89 \pm 0.02$ | $\mathbf{0.07 \pm 0.03}$ | $0.80 \pm 0.07$ | $\mathbf{36.90 \pm 11.73}$ | $\mathbf{0.85 \pm 0.05}$ |
| GOLEM | $\mathbf{0.04 \pm 0.04}$ | $\mathbf{0.97 \pm 0.02}$ | $\mathbf{8.10 \pm 6.98}$ | $\mathbf{0.97 \pm 0.03}$ | $0.15 \pm 0.06$ | $0.84 \pm 0.03$ | $36.00 \pm 11.81$ | $0.84 \pm 0.03$ |
| +**VISTA-NV** | $0.76 \pm 0.03$ | $0.93 \pm 0.03$ | $431.60 \pm 63.02$ | $0.38 \pm 0.04$ | $0.75 \pm 0.04$ | $\mathbf{0.93 \pm 0.03}$ | $397.70 \pm 68.22$ | $0.40 \pm 0.05$ |
| +**VISTA-WV** | $0.12 \pm 0.05$ | $0.76 \pm 0.04$ | $47.90 \pm 9.80$ | $0.81 \pm 0.04$ | $\mathbf{0.09 \pm 0.11}$ | $0.85 \pm 0.08$ | $\mathbf{20.20 \pm 16.75}$ | $\mathbf{0.88 \pm 0.07}$ |
| DAG-GNN | $0.14 \pm 0.08$ | $0.86 \pm 0.14$ | $38.80 \pm 26.22$ | $0.86 \pm 0.08$ | $0.26 \pm 0.10$ | $0.73 \pm 0.11$ | $75.40 \pm 24.08$ | $0.73 \pm 0.07$ |
| +**VISTA-NV** | $0.73 \pm 0.02$ | $\mathbf{0.98 \pm 0.00}$ | $380.00 \pm 48.64$ | $0.42 \pm 0.03$ | $0.73 \pm 0.03$ | $\mathbf{0.98 \pm 0.00}$ | $378.00 \pm 50.22$ | $0.42 \pm 0.04$ |
| +**VISTA-WV** | $\mathbf{0.07 \pm 0.04}$ | $0.84 \pm 0.02$ | $\mathbf{29.30 \pm 2.05}$ | $\mathbf{0.89 \pm 0.01}$ | $\mathbf{0.05 \pm 0.03}$ | $0.84 \pm 0.05$ | $\mathbf{41.00 \pm 8.01}$ | $\mathbf{0.89 \pm 0.03}$ |
| CAM | —— | —— | —— | —— | —— | —— | —— | —— |
| +**VISTA-NV** | $0.87 \pm 0.02$ | $\mathbf{0.66 \pm 0.06}$ | $641.30 \pm 67.62$ | $0.22 \pm 0.03$ | $0.86 \pm 0.02$ | $\mathbf{0.71 \pm 0.05}$ | $611.20 \pm 64.82$ | $0.24 \pm 0.03$ |
| +**VISTA-WV** | $\mathbf{0.66 \pm 0.05}$ | $0.51 \pm 0.07$ | $\mathbf{192.00 \pm 34.23}$ | $\mathbf{0.40 \pm 0.05}$ | $\mathbf{0.65 \pm 0.06}$ | $0.51 \pm 0.10$ | $\mathbf{181.80 \pm 21.12}$ | $\mathbf{0.41 \pm 0.07}$ |
| GraN-DAG | $0.74 \pm 0.32$ | $0.09 \pm 0.04$ | $209.00 \pm 54.45$ | $0.11 \pm 0.05$ | $0.34 \pm 0.42$ | $0.08 \pm 0.04$ | $166.60 \pm 42.38$ | $0.12 \pm 0.03$ |
| +**VISTA-NV** | $0.67 \pm 0.15$ | $\mathbf{0.31 \pm 0.06}$ | $158.80 \pm 33.13$ | $0.32 \pm 0.08$ | $0.48 \pm 0.30$ | $\mathbf{0.34 \pm 0.09}$ | $195.20 \pm 29.46$ | $\mathbf{0.41 \pm 0.11}$ |
| +**VISTA-WV** | $\mathbf{0.29 \pm 0.08}$ | $0.26 \pm 0.05$ | $\mathbf{123.40 \pm 15.51}$ | $\mathbf{0.38 \pm 0.05}$ | $\mathbf{0.22 \pm 0.21}$ | $0.20 \pm 0.09$ | $\mathbf{118.80 \pm 23.75}$ | $0.32 \pm 0.12$ |
| SCORE | $0.69 \pm 0.05$ | $0.67 \pm 0.08$ | $166.20 \pm 59.57$ | $0.42 \pm 0.03$ | $0.64 \pm 0.06$ | $0.64 \pm 0.10$ | $115.30 \pm 31.50$ | $0.45 \pm 0.02$ |
| +**VISTA-NV** | $0.86 \pm 0.06$ | $\mathbf{0.95 \pm 0.03}$ | $980.80 \pm 79.12$ | $0.24 \pm 0.09$ | $0.90 \pm 0.04$ | $\mathbf{0.91 \pm 0.05}$ | $923.70 \pm 146.32$ | $0.18 \pm 0.07$ |
| +**VISTA-WV** | $\mathbf{0.33 \pm 0.04}$ | $0.74 \pm 0.02$ | $\mathbf{56.40 \pm 19.95}$ | $\mathbf{0.70 \pm 0.02}$ | $\mathbf{0.49 \pm 0.40}$ | $0.80 \pm 0.06$ | $\mathbf{74.60 \pm 22.43}$ | $\mathbf{0.62 \pm 0.30}$ |

Table 12: Results with linear and nonlinear synthetic datasets ($n = 50$, $h = 5$).

| Method | ER5 | | | | SF5 | | | |
| --- | --- | --- | --- | --- | --- | --- | --- | --- |
| | FDR↓ | TPR↑ | SHD↓ | F1↑ | FDR↓ | TPR↑ | SHD↓ | F1↑ |
| NOTEARS | $0.16 \pm 0.09$ | $0.81 \pm 0.06$ | $81.80 \pm 34.23$ | $0.82 \pm 0.05$ | $0.23 \pm 0.08$ | $0.75 \pm 0.04$ | $105.90 \pm 26.65$ | $0.76 \pm 0.04$ |
| +**VISTA-NV** | $0.75 \pm 0.02$ | $\mathbf{0.98 \pm 0.01}$ | $685.60 \pm 68.09$ | $0.40 \pm 0.02$ | $0.72 \pm 0.02$ | $\mathbf{0.98 \pm 0.01}$ | $585.30 \pm 71.56$ | $0.43 \pm 0.03$ |
| +**VISTA-WV** | $\mathbf{0.08 \pm 0.04}$ | $0.76 \pm 0.04$ | $\mathbf{72.90 \pm 13.75}$ | $\mathbf{0.83 \pm 0.03}$ | $\mathbf{0.15 \pm 0.06}$ | $0.82 \pm 0.04$ | $\mathbf{71.90 \pm 12.54}$ | $\mathbf{0.84 \pm 0.02}$ |
| GOLEM | $0.34 \pm 0.18$ | $0.73 \pm 0.14$ | $156.20 \pm 88.04$ | $0.72 \pm 0.14$ | $0.30 \pm 0.17$ | $0.75 \pm 0.12$ | $130.20 \pm 71.76$ | $0.72 \pm 0.11$ |
| +**VISTA-NV** | $0.75 \pm 0.02$ | $\mathbf{0.96 \pm 0.02}$ | $706.90 \pm 66.89$ | $0.39 \pm 0.02$ | $0.74 \pm 0.03$ | $\mathbf{0.94 \pm 0.01}$ | $618.90 \pm 91.19$ | $0.41 \pm 0.04$ |
| +**VISTA-WV** | $\mathbf{0.20 \pm 0.11}$ | $0.77 \pm 0.08$ | $\mathbf{99.40 \pm 47.64}$ | $\mathbf{0.79 \pm 0.10}$ | $\mathbf{0.16 \pm 0.09}$ | $0.77 \pm 0.06$ | $\mathbf{84.40 \pm 34.04}$ | $\mathbf{0.80 \pm 0.08}$ |
| DAG-GNN | $0.29 \pm 0.15$ | $0.70 \pm 0.17$ | $141.10 \pm 63.30$ | $0.71 \pm 0.11$ | $0.32 \pm 0.11$ | $0.74 \pm 0.07$ | $142.40 \pm 45.01$ | $0.71 \pm 0.07$ |
| +**VISTA-NV** | $0.76 \pm 0.01$ | $\mathbf{0.98 \pm 0.01}$ | $720.70 \pm 37.85$ | $0.38 \pm 0.02$ | $0.74 \pm 0.01$ | $\mathbf{0.98 \pm 0.01}$ | $633.00 \pm 39.65$ | $0.41 \pm 0.01$ |
| +**VISTA-WV** | $\mathbf{0.22 \pm 0.07}$ | $0.79 \pm 0.04$ | $\mathbf{99.00 \pm 26.95}$ | $\mathbf{0.79 \pm 0.05}$ | $\mathbf{0.27 \pm 0.05}$ | $0.76 \pm 0.03$ | $\mathbf{116.60 \pm 11.84}$ | $\mathbf{0.75 \pm 0.01}$ |
| CAM | —— | —— | —— | —— | —— | —— | —— | —— |
| +**VISTA-NV** | $0.86 \pm 0.01$ | $\mathbf{0.69 \pm 0.05}$ | $978.00 \pm 5.25$ | $0.23 \pm 0.02$ | $0.86 \pm 0.01$ | $\mathbf{0.67 \pm 0.06}$ | $941.60 \pm 50.21$ | $0.23 \pm 0.02$ |
| +**VISTA-WV** | $\mathbf{0.75 \pm 0.02}$ | $0.49 \pm 0.08$ | $\mathbf{426.40 \pm 26.82}$ | $\mathbf{0.32 \pm 0.03}$ | $\mathbf{0.75 \pm 0.03}$ | $0.47 \pm 0.08$ | $\mathbf{400.80 \pm 41.11}$ | $\mathbf{0.33 \pm 0.05}$ |
| GraN-DAG | $0.62 \pm 0.28$ | $0.05 \pm 0.03$ | $265.80 \pm 35.61$ | $0.08 \pm 0.04$ | $0.64 \pm 0.33$ | $0.07 \pm 0.05$ | $271.80 \pm 29.98$ | $0.10 \pm 0.06$ |
| +**VISTA-NV** | $0.51 \pm 0.08$ | $\mathbf{0.17 \pm 0.09}$ | $213.40 \pm 82.48$ | $\mathbf{0.25 \pm 0.10}$ | $0.56 \pm 0.15$ | $\mathbf{0.26 \pm 0.05}$ | $229.20 \pm 63.68$ | $\mathbf{0.32 \pm 0.06}$ |
| +**VISTA-WV** | $\mathbf{0.36 \pm 0.05}$ | $0.13 \pm 0.02$ | $\mathbf{204.00 \pm 47.76}$ | $0.22 \pm 0.03$ | $\mathbf{0.27 \pm 0.05}$ | $0.18 \pm 0.04$ | $\mathbf{193.00 \pm 57.21}$ | $0.29 \pm 0.05$ |
| SCORE | $0.73 \pm 0.05$ | $\mathbf{0.61 \pm 0.15}$ | $431.60 \pm 114.55$ | $0.34 \pm 0.02$ | $0.71 \pm 0.04$ | $0.42 \pm 0.04$ | $365.00 \pm 68.00$ | $0.34 \pm 0.03$ |
| +**VISTA-NV** | $0.84 \pm 0.01$ | $0.47 \pm 0.32$ | $686.00 \pm 62.50$ | $0.24 \pm 0.04$ | $0.81 \pm 0.03$ | $\mathbf{0.54 \pm 0.06}$ | $582.50 \pm 57.50$ | $0.28 \pm 0.03$ |
| +**VISTA-WV** | $\mathbf{0.64 \pm 0.07}$ | $0.38 \pm 0.06$ | $\mathbf{271.00 \pm 43.00}$ | $\mathbf{0.37 \pm 0.05}$ | $\mathbf{0.35 \pm 0.15}$ | $0.25 \pm 0.04$ | $\mathbf{210.50 \pm 11.50}$ | $\mathbf{0.36 \pm 0.04}$ |

Table 13: Results with linear and nonlinear synthetic datasets ($n = 100$, $h = 3$).

| Method | ER3 | | | | SF3 | | | |
| --- | --- | --- | --- | --- | --- | --- | --- | --- |
| | FDR↓ | TPR↑ | SHD↓ | F1↑ | FDR↓ | TPR↑ | SHD↓ | F1↑ |
| NOTEARS | $\mathbf{0.09 \pm 0.09}$ | $0.91 \pm 0.06$ | $\mathbf{54.60 \pm 44.78}$ | $\mathbf{0.91 \pm 0.08}$ | $0.15 \pm 0.08$ | $0.75 \pm 0.06$ | $108.80 \pm 36.84$ | $0.80 \pm 0.06$ |
| +**VISTA-NV** | $0.81 \pm 0.04$ | $\mathbf{0.95 \pm 0.01}$ | $1245.60 \pm 349.77$ | $0.32 \pm 0.05$ | $0.75 \pm 0.04$ | $\mathbf{0.95 \pm 0.03}$ | $864.00 \pm 146.07$ | $0.39 \pm 0.05$ |
| +**VISTA-WV** | $0.09 \pm 0.02$ | $0.73 \pm 0.02$ | $99.00 \pm 3.77$ | $0.81 \pm 0.01$ | $\mathbf{0.11 \pm 0.03}$ | $0.80 \pm 0.05$ | $\mathbf{92.00 \pm 9.67}$ | $\mathbf{0.85 \pm 0.03}$ |
| GOLEM | $\mathbf{0.09 \pm 0.10}$ | $\mathbf{0.95 \pm 0.05}$ | $39.80 \pm 43.52$ | $\mathbf{0.93 \pm 0.08}$ | $0.22 \pm 0.05$ | $0.72 \pm 0.04$ | $137.00 \pm 22.24$ | $0.75 \pm 0.04$ |
| +**VISTA-NV** | $0.84 \pm 0.01$ | $0.91 \pm 0.02$ | $1373.80 \pm 202.28$ | $0.28 \pm 0.02$ | $0.81 \pm 0.03$ | $\mathbf{0.90 \pm 0.02}$ | $1180.20 \pm 163.44$ | $0.31 \pm 0.04$ |
| +**VISTA-WV** | $0.22 \pm 0.02$ | $0.65 \pm 0.04$ | $147.60 \pm 17.55$ | $0.71 \pm 0.03$ | $\mathbf{0.18 \pm 0.13}$ | $0.78 \pm 0.06$ | $\mathbf{91.80 \pm 72.13}$ | $\mathbf{0.80 \pm 0.07}$ |
| DAG-GNN | $0.15 \pm 0.11$ | $0.71 \pm 0.17$ | $119.40 \pm 63.78$ | $0.77 \pm 0.15$ | $0.31 \pm 0.14$ | $0.54 \pm 0.09$ | $215.60 \pm 47.23$ | $0.59 \pm 0.06$ |
| +**VISTA-NV** | $0.63 \pm 0.03$ | $\mathbf{0.95 \pm 0.01}$ | $1239.60 \pm 131.07$ | $0.53 \pm 0.03$ | $0.78 \pm 0.04$ | $\mathbf{0.94 \pm 0.01}$ | $1058.40 \pm 259.74$ | $0.34 \pm 0.06$ |
| +**VISTA-WV** | $\mathbf{0.12 \pm 0.02}$ | $0.82 \pm 0.03$ | $\mathbf{87.20 \pm 15.30}$ | $\mathbf{0.85 \pm 0.02}$ | $\mathbf{0.23 \pm 0.10}$ | $0.70 \pm 0.08$ | $\mathbf{151.20 \pm 25.35}$ | $\mathbf{0.73 \pm 0.03}$ |
| GraN-DAG | $0.90 \pm 0.03$ | $0.04 \pm 0.02$ | $463.40 \pm 22.94$ | $0.04 \pm 0.02$ | $0.83 \pm 0.07$ | $0.02 \pm 0.02$ | $366.40 \pm 118.26$ | $0.05 \pm 0.02$ |
| +**VISTA-NV** | $0.88 \pm 0.06$ | $\mathbf{0.25 \pm 0.06}$ | $390.80 \pm 73.58$ | $0.17 \pm 0.06$ | $0.78 \pm 0.06$ | $\mathbf{0.26 \pm 0.08}$ | $308.40 \pm 49.60$ | $0.24 \pm 0.05$ |
| +**VISTA-WV** | $\mathbf{0.38 \pm 0.05}$ | $0.16 \pm 0.03$ | $\mathbf{250.60 \pm 82.64}$ | $\mathbf{0.25 \pm 0.04}$ | $\mathbf{0.44 \pm 0.08}$ | $0.18 \pm 0.05$ | $\mathbf{266.68 \pm 67.76}$ | $\mathbf{0.27 \pm 0.06}$ |
| SCORE | $0.91 \pm 0.05$ | $0.62 \pm 0.04$ | $2859.40 \pm 839.4$ | $0.16 \pm 0.08$ | $0.92 \pm 0.03$ | $0.66 \pm 0.04$ | $3131.20 \pm 1002$ | $0.14 \pm 0.05$ |
| +**VISTA-NV** | $0.94 \pm 0.04$ | $\mathbf{0.95 \pm 0.12}$ | $2614.80 \pm 566.5$ | $0.11 \pm 0.07$ | $0.91 \pm 0.05$ | $\mathbf{0.70 \pm 0.05}$ | $2727.60 \pm 505.6$ | $0.16 \pm 0.08$ |
| +**VISTA-WV** | $\mathbf{0.53 \pm 0.05}$ | $0.75 \pm 0.06$ | $\mathbf{339.00 \pm 189.43}$ | $\mathbf{0.58 \pm 0.04}$ | $\mathbf{0.51 \pm 0.10}$ | $0.68 \pm 0.08$ | $\mathbf{408.80 \pm 205.64}$ | $\mathbf{0.57 \pm 0.07}$ |

Table 14: Results with linear and nonlinear synthetic datasets ($n = 300$, $h = 3$).

| Method | ER3 | | | | SF3 | | | |
| --- | --- | --- | --- | --- | --- | --- | --- | --- |
| | FDR↓ | TPR↑ | SHD↓ | F1↑ | FDR↓ | TPR↑ | SHD↓ | F1↑ |
| NOTEARS | $\mathbf{0.13 \pm 0.03}$ | $0.89 \pm 0.02$ | $\mathbf{202.33 \pm 48.98}$ | $\mathbf{0.88 \pm 0.03}$ | $\mathbf{0.16 \pm 0.06}$ | $0.72 \pm 0.04$ | $519.33 \pm 71.15$ | $\mathbf{0.78 \pm 0.04}$ |
| +**VISTA-NV** | $0.88 \pm 0.01$ | $\mathbf{0.91 \pm 0.00}$ | $6177.00 \pm 372.80$ | $0.21 \pm 0.02$ | $0.89 \pm 0.00$ | $\mathbf{0.91 \pm 0.01}$ | $6917.67 \pm 998.81$ | $0.20 \pm 0.00$ |
| +**VISTA-WV** | $0.23 \pm 0.02$ | $0.66 \pm 0.03$ | $462.00 \pm 54.31$ | $0.71 \pm 0.02$ | $0.21 \pm 0.03$ | $0.55 \pm 0.03$ | $\mathbf{363.00 \pm 76.53}$ | $0.65 \pm 0.02$ |
| GOLEM | $0.23 \pm 0.15$ | $0.76 \pm 0.18$ | $\mathbf{375.67 \pm 258.67}$ | $\mathbf{0.77 \pm 0.16}$ | $0.56 \pm 0.19$ | $0.34 \pm 0.22$ | $913.33 \pm 304.25$ | $0.38 \pm 0.21$ |
| +**VISTA-NV** | $0.88 \pm 0.00$ | $\mathbf{0.85 \pm 0.01}$ | $5389.67 \pm 46.91$ | $0.22 \pm 0.00$ | $0.86 \pm 0.03$ | $\mathbf{0.48 \pm 0.28}$ | $3248.67 \pm 1304.26$ | $0.18 \pm 0.08$ |
| +**VISTA-WV** | $\mathbf{0.17 \pm 0.02}$ | $0.45 \pm 0.01$ | $628.33 \pm 11.90$ | $0.50 \pm 0.01$ | $\mathbf{0.21 \pm 0.06}$ | $0.44 \pm 0.04$ | $\mathbf{597.00 \pm 53.59}$ | $\mathbf{0.56 \pm 0.04}$ |
| DAG-GNN | $0.55 \pm 0.34$ | $0.19 \pm 0.19$ | $1288.33 \pm 832.49$ | $0.21 \pm 0.20$ | $0.72 \pm 0.17$ | $0.23 \pm 0.22$ | $1264.33 \pm 484.00$ | $0.22 \pm 0.19$ |
| +**VISTA-NV** | $0.89 \pm 0.01$ | $\mathbf{0.93 \pm 0.00}$ | $6449.00 \pm 89.16$ | $0.19 \pm 0.01$ | $0.89 \pm 0.01$ | $\mathbf{0.89 \pm 0.02}$ | $6627.00 \pm 651.98$ | $0.19 \pm 0.02$ |
| +**VISTA-WV** | $\mathbf{0.18 \pm 0.06}$ | $0.57 \pm 0.07$ | $\mathbf{494.67 \pm 50.22}$ | $\mathbf{0.66 \pm 0.04}$ | $\mathbf{0.34 \pm 0.06}$ | $0.49 \pm 0.07$ | $\mathbf{633.33 \pm 111.52}$ | $\mathbf{0.55 \pm 0.03}$ |
| SCORE | —— | —— | —— | —— | —— | —— | —— | —— |
| +**VISTA-NV** | $0.95 \pm 0.00$ | $\mathbf{0.76 \pm 0.04}$ | $11064.00 \pm 371.63$ | $0.09 \pm 0.01$ | $0.97 \pm 0.01$ | $\mathbf{0.44 \pm 0.13}$ | $13057.00 \pm 3556.57$ | $0.06 \pm 0.02$ |
| +**VISTA-WV** | $\mathbf{0.19 \pm 0.02}$ | $0.32 \pm 0.03$ | $\mathbf{666.67 \pm 18.66}$ | $\mathbf{0.46 \pm 0.03}$ | $\mathbf{0.61 \pm 0.29}$ | $0.08 \pm 0.04$ | $\mathbf{970.67 \pm 141.74}$ | $\mathbf{0.13 \pm 0.07}$ |

Table 15: Results with linear and nonlinear synthetic datasets ($n = 300$, $h = 5$).

| Method | ER5 | | | | SF5 | | | |
|---|---|---|---|---|---|---|---|---|
| | FDR↓ | TPR↑ | SHD↓ | F1↑ | FDR↓ | TPR↑ | SHD↓ | F1↑ |
| NOTEARS | $0.30 \pm 0.05$ | $0.68 \pm 0.12$ | $875.33 \pm 205.07$ | $0.69 \pm 0.09$ | $0.50 \pm 0.09$ | $0.22 \pm 0.12$ | $1402.33 \pm 70.59$ | $0.29 \pm 0.14$ |
| +**VISTA-NV** | $0.93 \pm 0.02$ | $\mathbf{0.94 \pm 0.01}$ | $6520.33 \pm 1357.12$ | $0.10 \pm 0.02$ | $0.90 \pm 0.01$ | $\mathbf{0.78 \pm 0.04}$ | $12180.00 \pm 1008.42$ | $0.18 \pm 0.02$ |
| +**VISTA-WV** | $\mathbf{0.15 \pm 0.04}$ | $0.67 \pm 0.05$ | $\mathbf{689.67 \pm 98.89}$ | $\mathbf{0.75 \pm 0.03}$ | $\mathbf{0.24 \pm 0.02}$ | $0.38 \pm 0.03$ | $\mathbf{890.33 \pm 166.61}$ | $\mathbf{0.51 \pm 0.03}$ |
| GOLEM | $0.81 \pm 0.05$ | $0.10 \pm 0.03$ | $1921.33 \pm 111.21$ | $0.13 \pm 0.04$ | $0.93 \pm 0.03$ | $0.02 \pm 0.01$ | $1839.67 \pm 139.17$ | $0.03 \pm 0.02$ |
| +**VISTA-NV** | $0.92 \pm 0.01$ | $\mathbf{0.28 \pm 0.14}$ | $5551.00 \pm 1310.12$ | $0.12 \pm 0.03$ | $0.92 \pm 0.00$ | $\mathbf{0.77 \pm 0.09}$ | $13437.00 \pm 1562.43$ | $0.14 \pm 0.00$ |
| +**VISTA-WV** | $\mathbf{0.20 \pm 0.23}$ | $0.23 \pm 0.02$ | $\mathbf{1225.00 \pm 38.79}$ | $\mathbf{0.36 \pm 0.03}$ | $\mathbf{0.37 \pm 0.11}$ | $0.10 \pm 0.06$ | $\mathbf{1391.00 \pm 61.65}$ | $\mathbf{0.17 \pm 0.10}$ |
| DAG-GNN | $0.91 \pm 0.06$ | $0.33 \pm 0.17$ | $3858.00 \pm 1558.37$ | $0.17 \pm 0.04$ | $0.91 \pm 0.04$ | $0.15 \pm 0.06$ | $4617.33 \pm 3064.50$ | $0.10 \pm 0.04$ |
| +**VISTA-NV** | $0.90 \pm 0.03$ | $\mathbf{0.86 \pm 0.04}$ | $8988.00 \pm 910.33$ | $0.18 \pm 0.05$ | $0.91 \pm 0.03$ | $\mathbf{0.81 \pm 0.04}$ | $14578.67 \pm 4342.92$ | $0.16 \pm 0.05$ |
| +**VISTA-WV** | $\mathbf{0.37 \pm 0.14}$ | $0.25 \pm 0.05$ | $\mathbf{1920.33 \pm 809.62}$ | $\mathbf{0.36 \pm 0.06}$ | $\mathbf{0.41 \pm 0.03}$ | $0.21 \pm 0.06$ | $\mathbf{2191.33 \pm 656.02}$ | $\mathbf{0.31 \pm 0.09}$ |
| SCORE | —— | —— | —— | —— | —— | —— | —— | —— |
| +**VISTA-NV** | $0.96 \pm 0.00$ | $\mathbf{0.17 \pm 0.07}$ | $18762.67 \pm 2501.28$ | $0.06 \pm 0.01$ | $0.98 \pm 0.00$ | $\mathbf{0.13 \pm 0.15}$ | $22039.00 \pm 2028.89$ | $0.03 \pm 0.01$ |
| +**VISTA-WV** | $\mathbf{0.93 \pm 0.00}$ | $0.10 \pm 0.03$ | $\mathbf{1698.33 \pm 103.76}$ | $\mathbf{0.07 \pm 0.01}$ | $\mathbf{0.95 \pm 0.02}$ | $0.08 \pm 0.06$ | $\mathbf{2582.67 \pm 830.34}$ | $\mathbf{0.06 \pm 0.02}$ |

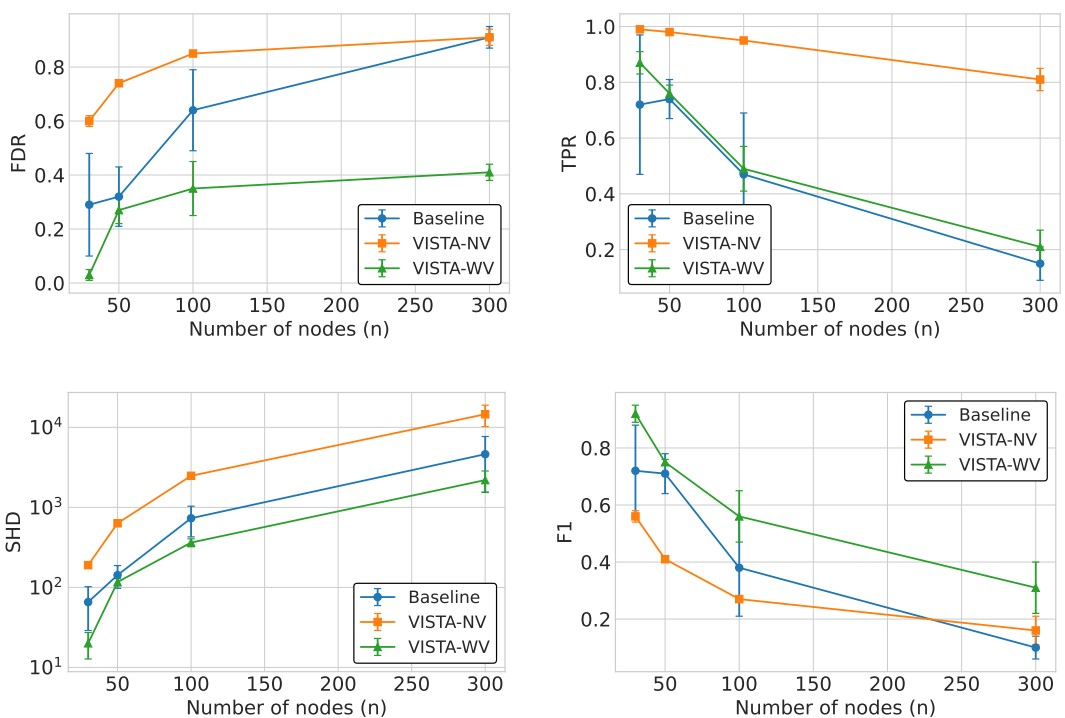

Figure 5: Performance of DAG-GNN on SF5 Graphs.

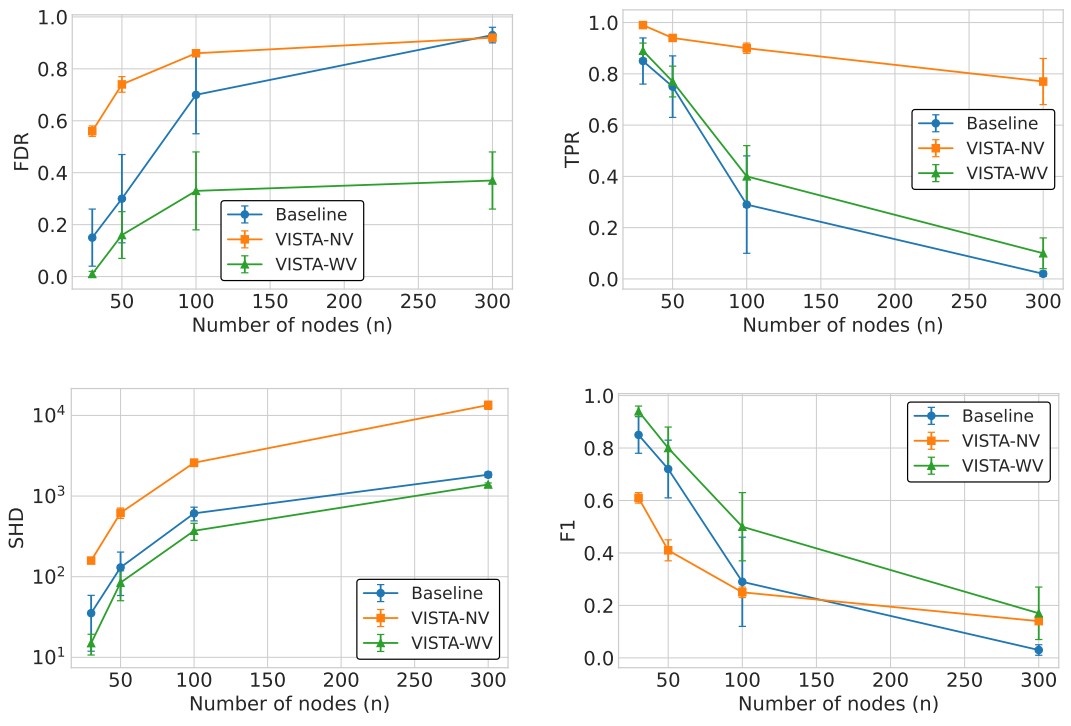

Figure 6: Performance of GOLEM on SF5 Graphs.

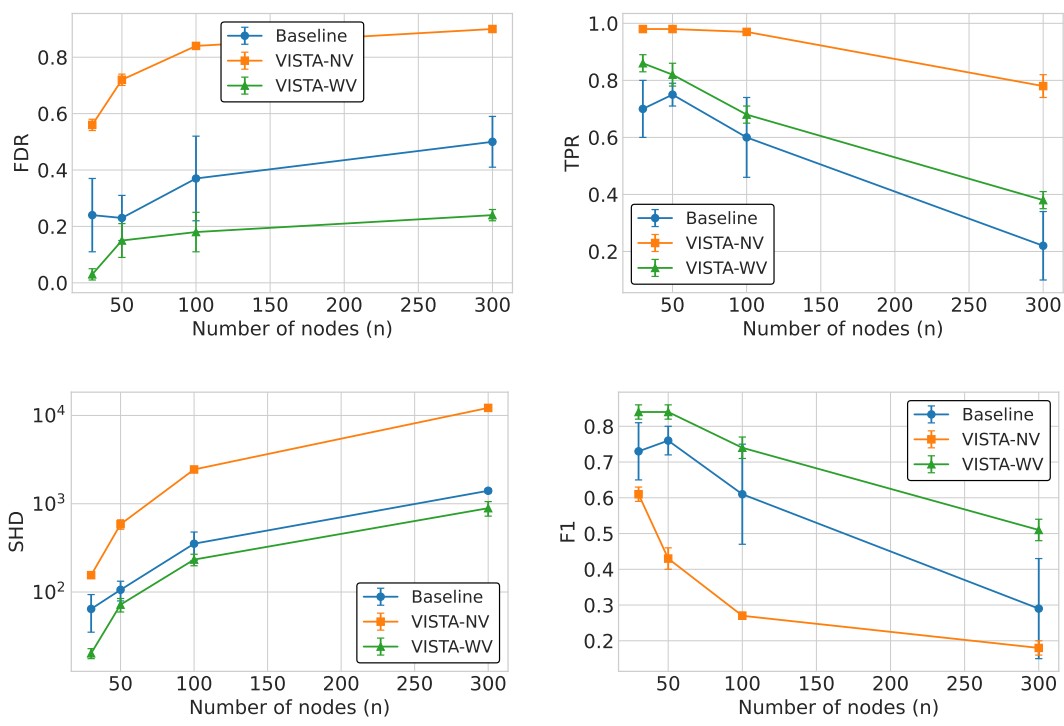

Figure 7: Performance of NOTEARS on SF5 Graphs.

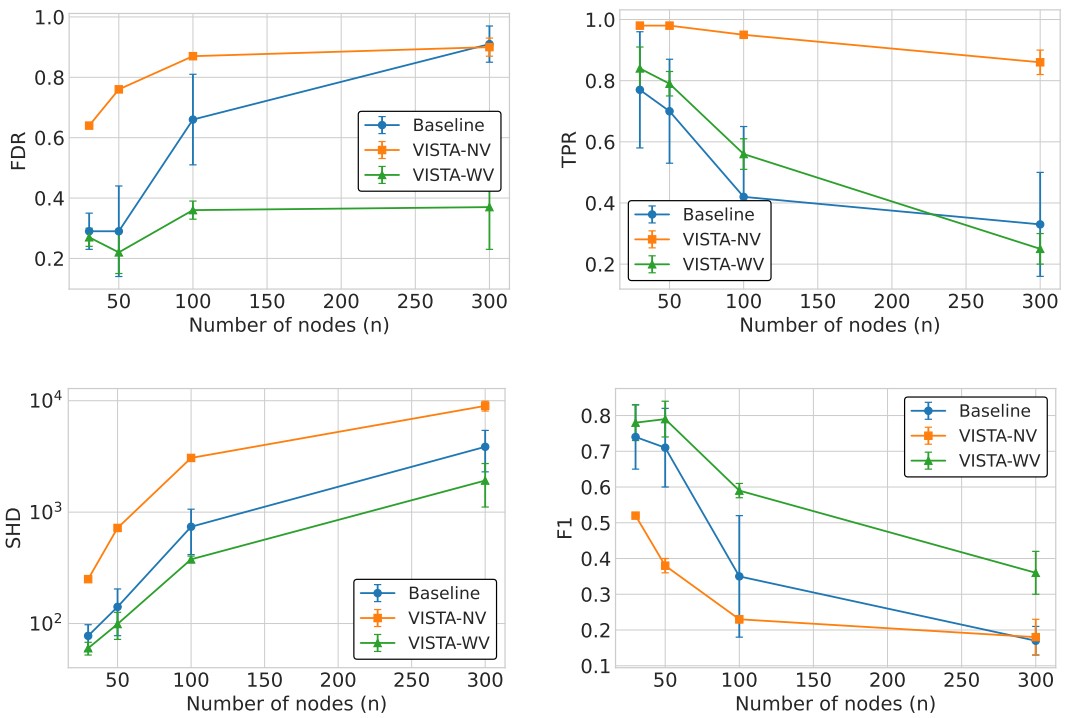

Figure 8: Performance of DAG-GNN on ER5 Graphs.

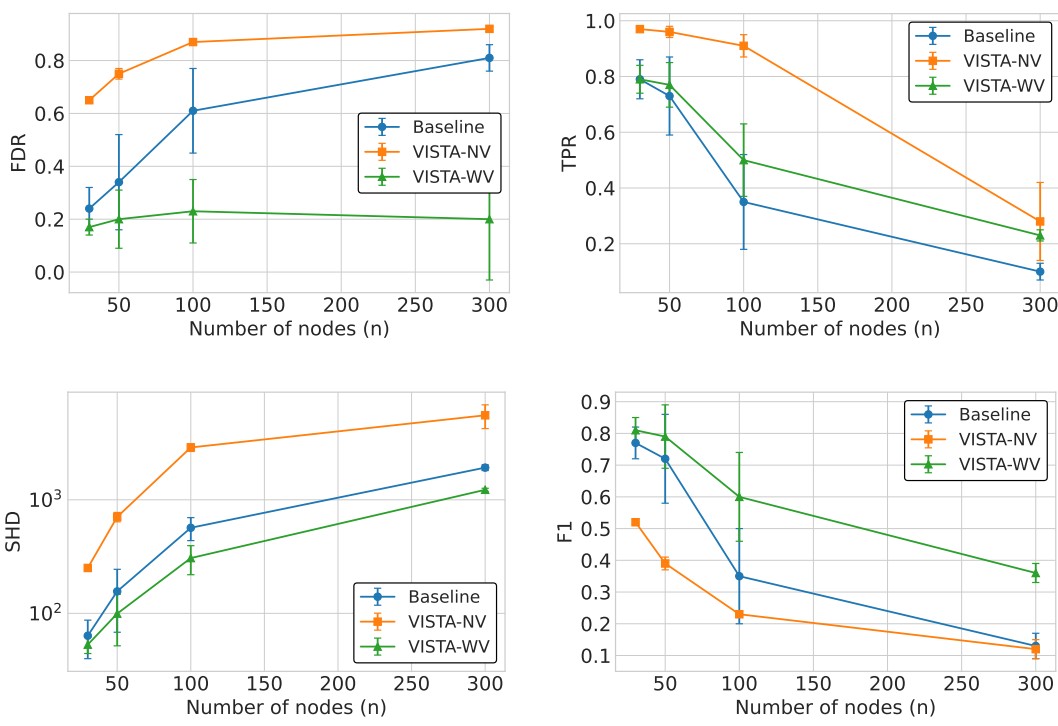

Figure 9: Performance of GOLEM on ER5 Graphs.

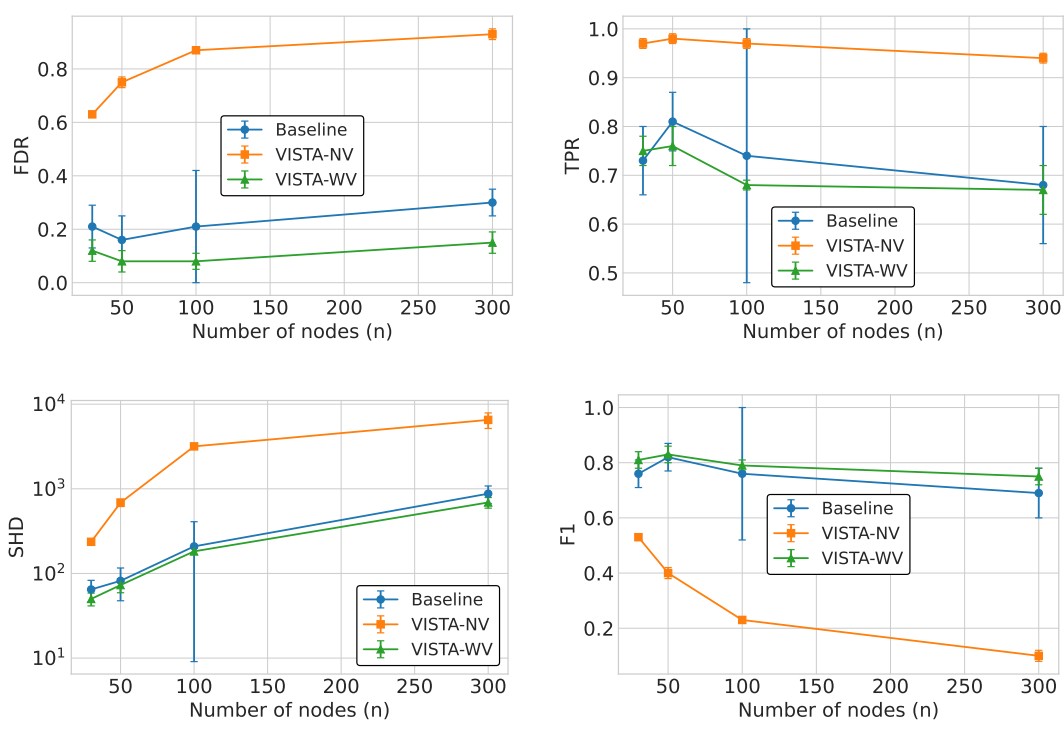

Figure 10: Performance of NOTEARS on ER5 Graphs.

