# OpenReview forum: "Efficient DAG Learning via Modular Subgraph Integration"
_TMLR — Under review for TMLR_

### Review · Reviewer_fq2V · 2026-03-16

**Summary Of Contributions:**

### Summary

This paper proposes causal discovery via a hybrid approach that identifies subgraphs, performs causal discovery on the subgraphs, then combines them with a voting procedure. In particular, the subgraphs are defined via the Markov blanket of each node; discovery on each Markov blanket can then occur independently and in parallel. To combine, two voting procedures are proposed: one in which voting occurs irrespective of the total number of times an edge is detected, and another which performs voting and thresholds based on how many times an edge was detected. Theory is presented about the reliability of such methods, and experiments which show several existing methods are improved when combined with this framework.

### Strengths

- The idea of using subgraphs and combining that information is nice; existing approaches often don't use parallelization as well as they could, and the VISTA framework provides a clear path to doing so.
- The experiments show impressive results in improving performance across a wide array of different algorithms.

### Weaknesses

Unfortunately, I have serious concerns about both theory and experiments. I state these below.

**Additional Comments:**

## References

\[1\] Reisach, A., Seiler, C., & Weichwald, S. (2021). Beware of the simulated dag! causal discovery benchmarks may be easy to game. _Advances in Neural Information Processing Systems_, _34_, 27772-27784.

\[2\] Göbler, K., Windisch, T., Drton, M., Pychynski, T., Roth, M., & Sonntag, S. (2024, March). $\texttt {causalAssembly} $: Generating Realistic Production Data for Benchmarking Causal Discovery. In _Causal Learning and Reasoning_ (pp. 609-642). PMLR.

**Audience:**

No

**Audience Explanation:**

The topic of the paper -- and the core idea of creating hybrid discovery algorithms with subgraphs via Markov blankets -- would be of interest to the TMLR audience. But due to the issues in writing and correctness, I don't think it the manuscript is, in its current state.

**Claims And Evidence:**

No

**Claims Explanation:**

There are several claims I find lacking in evidence.

**\[MB Performance vs $d$\]** It's stated "Besides, as shown in Figure 1, the accuracy of MB identification remains relatively stable as the number of nodes increases [...]". But there are no details about how Figure 1 was generated, in the text or in the caption. This could depend on many different factors, e.g., amount of data, the type of graph, and the MB learner. I also don't quite understand what it means to compare the F1 score of "Markov Blanket" and (NOTEARS) in the same plot.

**\[Definition of NV Scheme\]** The section on naïve voting stops short of actually writing a voting mechanism. In particular, it defines $r_{X\to Y}$ and $r_{Y\to X}$, but nothing about the corresponding decision rule. For example, does one sample with probabilities $A/(A+B), B/(A+B)$? Or just tax the maximum (in which case, why divide by $A+B$)?

**\[Model Agnostic Claims\]** A core claim of the paper is that VISTA is model agnostic, which I think is too strong. For example, "if an undirected adjacency $X - Y$ is returned, it is treated as providing no directional vote in the aggregation" -- but this is not uncommon in observational CD algorithms, and indeed carries significant causal information in many methods.

**\[Experiment Hyperparameter Choice\]** There are some lacking details in the experimental section.
1. It's written that "all VISTA results in the main tables use a single, fixed oeprating point: $\lambda = 0.5$ [...] this choice lies within (5)". But (5) specifies an upper bound depending on $\epsilon$, which is not specified.
2. How Markov blankets are learned is not specified.

**\[Experiment Datasets\]** It's become quite well-known recently that synthetic datasets are problematic for benchmarking causal discovery (c.f., \[1\]). To this extent, only the Sachs dataset is presented as a real dataset, and its results are much less conclusive. It would benefit the paper to include experiments on more recent causal discovery benchmarks, e.g., from `causalAssembly` \[2\].

**\[Theorems\]** The theorems seem to have some issues. I did not read proofs in detail, but spotted these two details from a cursory reading:
1. For example, Theorem 3.4 says that $\lambda \leq - \ln \epsilon / m$ is admissible for achieving error rate $\epsilon$. But this implies that the admissible range of $\lambda$ actually *increases* as the error bound gets *tighter*.
2. The proof of corollary 3.3 also takes a Taylor expansion then simply drops the second order term, with no justification other than that it is "small enough." This is not sufficient to rigorously prove an inequality.

**Requested Changes:**

### Evidence

Please clarify or address the points noted in Evidence. In particular, \[MB Performance vs $d$\], \[Definition of NV Scheme\], \[Model Agnostic Claims\], \[Experiment Hyperparameter Choice\], \[Experiment Datasets\], and \[Theorems\].

### Editorial Remarks

In addition to claims and evidence, I have several editorial remarks that should be addressed.

**\[Citation Formatting\]** There are many places (e.g., the entire introduction) where `\citet{...}` (or `\cite{...}`) are used but should actually be parenthetical citations (`\citep{...}`).

**\[Using Unintroduced Concepts\]** There are a few places where concepts are used before they are introduced in the main text. For example:
1. On page 1, "while identifiability of the true DAG generally requires additional structural assumptions, our VISTA framework [...]" comes before VISTA is ever introduced.
2. One page 2, "[VISTA] operates purely on the edge-level outputs of local subgraphs and requires only a one-time O(|V |2) aggregation without any additional solver or training overhead." $V$ is not introduced.
	1. This is further confusing as $d$ is then used (again without introduction) on page 3.
3. On page 4, discussions of experimental results of VISTA already start ("furthermore, across different graph sizes, the VISTA-enhanced versions consistently outperform their corresponding baselines [...]"), again before VISTA is actually introduced.

**\[Minor Factual Issues\]** There are a few minor issues with claims:
1. Page 2, "NOTEARS [..., DAGMA, ...] convert acyclicity into a smooth penalty and learn graphs via gradient descent." These methods do not use gradient descent: NOTEARS uses an augmented Lagrangian optimizer, and DAGMA uses a barrier method.
2. Page 4, "A structural causal model with mutually independent noises $\epsilon_i$: $V_i = f_i(\mathrm{Pa}(V_i), \epsilon_i), \epsilon_i \perp \mathrm{Pa}(V_i).$" This mathematical statement that follows does not state that the noises are mutually independent.
3. Page 4, "This induces a directed acyclic graph [...]" this model does not induce a DAG -- the definition uses the DAG already by referring to $\mathrm{Pa}(V_i)$.

**\[Typos\]** I caught a few typos:
1. On page 2, "There are also works that characterizes intervnetional" mispells "interventional".
2. On page 5, I think the period in "however, while NV does not distinguish between strong and weak statistical support. Edges appearing [...]" is supposed to be a comma.

---

> ### Author Response · Authors · 2026-06-11
> **Reply to Reviewer fq2V (Part 1 of 3)**
>
> We thank the reviewer for the careful reading and constructive suggestions. Below, we respond to each comment point by point and summarize the corresponding clarifications and revisions made in the manuscript.
>
> > It's stated "Besides, as shown in Figure 1, the accuracy of MB identification remains relatively stable as the number of nodes increases [...]". But there are no details about how Figure 1 was generated, in the text or in the caption. This could depend on many different factors, e.g., amount of data, the type of graph, and the MB learner. I also don't quite understand what it means to compare the F1 score of "Markov Blanket" and (NOTEARS) in the same plot.
>
> Thanks for pointing this out. The purpose of Figure 1 is not to compare Markov Blanket identification with full DAG learning as two equivalent causal discovery methods. Rather, Figure 1 is intended as a diagnostic edge-level comparison to illustrate why the MB-based decomposition is useful as graph size increases. All methods in this figure are evaluated at the skeleton level (and directions ignored). Specifically, an estimated Markov Blanket induces a set of candidate adjacencies, and a candidate adjacency is counted as correct if the corresponding undirected edge exists in the true graph. The NOTEARS and DAG-GNN curves are also converted to skeleton-level predictions before computing F1, so the comparison is performed at the same edge-level resolution rather than at the full directed-graph level.
>
> The main observation is that, as the number of nodes increases, the edge-identification accuracy of global base learners degrades more rapidly, while the MB-induced candidate skeleton remains relatively stable. This supports the motivation of VISTA: MB-based decomposition preserves a useful high-recall local candidate structure, and the subsequent voting and pruning stages then refine this candidate set into a cleaner DAG. To avoid ambiguity, we have revised the surrounding text of Figure 1.
>
>
> > The section on naive voting stops short of actually writing a voting mechanism. In particular, it defines $r_{X\to Y}$ and $r_{Y\to X}$, but says nothing about the corresponding decision rule. For example, does one sample with probabilities $A/(A+B)$ and $B/(A+B)$? Or just take the maximum, in which case why divide by $A+B$?
>
> Thanks for the comment. We would like to clarify that naive voting is a deterministic majority-vote rule. For each unordered pair $\{X,Y\}$, let $A$ be the number of local subgraphs supporting $X\to Y$, and $B$ the number supporting $Y\to X$. If $A+B>0$, NV selects the orientation with larger support: $X\to Y$ if $A>B$, and $Y\to X$ if $B>A$. In the rare case $A=B$, the pair is treated as unresolved and no directed edge is added by NV. The ratios $A/(A+B)$ and $B/(A+B)$ are used only as normalized directional support scores, so that the two orientations can be compared on a common scale and to make the relationship between NV and the weighted-voting rule clearer. We have revised the NV subsection to explicitly state this decision rule.

---

> ### Author Response · Authors · 2026-06-11
> **Reply to Reviewer fq2V (Part 2 of 3)**
>
> > A core claim of the paper is that VISTA is model agnostic, which I think is too strong. For example, "if an undirected adjacency $C-D$ is returned, it is treated as providing no directional vote in the aggregation" -- but this is not uncommon in observational CD algorithms, and indeed carries significant causal information in many methods.
>
> In the revised manuscript, we have relaxed this claim and clarified our intended scope is only within DAG learning. VISTA is agnostic to the internal objective, parametric form, and optimization procedure of the base learner within the DAG-learning setting, provided that the local learner outputs directed edges or directed edge scores that can be aggregated by our voting rule. We have also revised the manuscript to state that extending VISTA to incorporate partially directed outputs or skeleton-level evidence is an important direction for future work.
>
>
>
>
> > It's written that "all VISTA results in the main tables use a single, fixed oeprating point: $\lambda=0.5$ [...] this choice lies within (5)". But (5) specifies an upper bound depending on $\epsilon$, which is not specified.
>
> We would like to clarify that $\epsilon$ in Eq. (5) is only a theoretical confidence parameter used only in the sufficient condition of Theorem 3.4. It is not an experimental hyperparameter and is not used by the VISTA algorithm in practice. This is analogous to the role of a target failure probability in PAC-style bounds: it specifies the desired confidence level of the theoretical guarantee, but it does not need to be tuned or selected when running the method.
>
> The experimental hyperparameters used by VISTA are only the voting weight $\lambda$ and the decision threshold $t$. In all main experiments, we fix $\lambda=0.5$ and $t=0.7$ without dataset-specific tuning. The statement that this choice is consistent with the theoretical range in Eq. (5) should be understood as a theoretical sanity check, not as an additional experimental tuning criterion. For example, if one takes a standard target confidence $1-\epsilon=0.95$, then $-\ln(\epsilon)\approx 3$, and the upper-bound term is easy to satisfy for the moderate vote counts considered in our local aggregation setting.
>
> To avoid this possible confusion, we have revised the experimental section to explicitly state that $\epsilon$ is only a theoretical confidence parameter and is not part of the experimental hyperparameter selection. We also added a short discussion clarifying that the fixed choice $\lambda=0.5$ and $t=0.7$ is used as a single operating point across all main tables, independent of any choice of $\epsilon$.
>
>
> > It's become quite well-known recently that synthetic datasets are problematic for benchmarking causal discovery (c.f., [1]). To this extent, only the Sachs dataset is presented as a real dataset, and its results are much less conclusive. It would benefit the paper to include experiments on more recent causal discovery benchmarks, e.g., from causalAssembly [2].
>
> We have added causalAssembly experiments in Section 4.3. We evaluate the same set of base learners, both in isolation and integrated with VISTA, under the same fixed aggregation parameters used in the rest of the paper. The new results are reported in Table 6. Overall, VISTA improves the structural recovery performance on this larger and more realistic benchmark, generally reducing FDR and SHD and improving F1 across the tested base learners.

---

> ### Author Response · Authors · 2026-06-11
> **Reply to Reviewer fq2V (Part 3 of 3)**
>
> > For example, Theorem 3.4 says that $\lambda \leq - \ln \epsilon /m$ is admissible for achieving error rate $\epsilon$. But this implies that the admissible range of $\lambda$ actually increases as the error bound gets tighter.
>
> As we replied earlier, the parameter $\epsilon$ in Eq. (5) is a theoretical confidence parameter controlling the exponential damping term, rather than an experimental hyperparameter or a directly tuned empirical error level. Since the upper endpoint is $-\ln(\epsilon)/m$, it indeed increases as $\epsilon$ decreases. This is not a contradiction in the intended sufficient condition: a smaller $\epsilon$ corresponds to requiring the exponential residual term to be smaller, which allows a larger admissible upper endpoint for $\lambda$. For example, $-\ln(0.05)\approx 3$, while $-\ln(0.01)\approx 4.6$. However, we agree that describing this interval as simply “achieving a tighter error bound” can be confusing. We have revised the statement and discussion of Theorem 3.4 to clarify the role of $\epsilon$.
>
>
> > How Markov blankets are learned is not specified.
>
> Thank you for the comment. We did state that we implemented the MB solver used in DCILP, the closest divide-and-conquer baseline. Following the suggestion from Reviewer Z7Xv, we have also moved the comparison with DCILP from the appendix to the main paper as Section 4.2. This makes the shared MB-based divide-and-conquer setting and the differences in the aggregation stage more visible.
>
>
> > The proof of corollary 3.3 also takes a Taylor expansion then simply drops the second order term, with no justification other than that it is "small enough." This is not sufficient to rigorously prove an inequality.
>
> In our setting, “small enough” refers to the fact that $y=\exp(-\lambda m)$ is already very small for the parameter range considered. For example, taking $\lambda=0.7$ and $m=5$, we have $y=\exp(-3.5)\approx 0.03$, and hence $y^2\approx 9\times 10^{-4}$. Thus, the second-order term contributes only at the level of $10^{-3}$, which is negligible for the desired accuracy. We agree that the original wording was not sufficiently rigorous, and we have revised the proof to explicitly state this approximation bound.
>
>
> > [Citation Formatting] and [Using Unintroduced Concepts] 1.2
>
> Thanks for the suggestion. We have now revised the format of citation and minor issues.
>
> > 3. On page 4, discussions of experimental results of VISTA already start ("furthermore, across different graph sizes, the VISTA-enhanced versions consistently outperform their corresponding baselines [...]"), again before VISTA is actually introduced.
>
> We would like to argue that we have formally introduced VISTA at the beginning of Section 3, while this part serves as a concise summary and empirical sanity check of the VISTA methodology. After checking the manuscript structure, we believe the current ordering is logically correct, so we did not make further changes to this part.
>
> > [Minor Factual Issues] and [Typos]
>
> Thanks. We have revised these minor issues.
>
>
> Taken together, we sincerely thank the reviewer for the careful reading and constructive comments. We hope that our responses and the corresponding revisions have addressed the reviewer’s concerns and improved the clarity, rigor, and empirical support of the paper.

---

### Review · Reviewer_Z7Xv · 2026-04-20

**Summary Of Contributions:**

This paper proposes VISTA, a modular framework for scalable causal structure learning that decomposes DAG discovery into Markov Blanket based local subproblems, applies an arbitrary base learner on each local subgraph, and then merges the local outputs through a weighted voting rule followed by a feedback arc set based acyclicity step. The paper also provides finite-sample style analysis for the weighted voting rule, an asymptotic consistency claim, and an experimental study across several synthetic settings, one real benchmark, and a comparison to DCILP.

**Strengths:**
1. The paper addresses an important and practically relevant problem, namely improving the scalability of causal discovery in high dimensions.
2. The framework is simple, modular, and broadly compatible with many base learners, which makes it potentially useful in practice.
3. The empirical study is fairly broad on synthetic benchmarks and suggests that the weighted voting variant can improve precision and runtime in many cases.

**Weaknesses:**
1. The strongest theoretical claims are not yet fully convincing, since the analysis relies on independence of local votes even though the local subgraphs overlap and are learned from the same dataset.
2. The asymptotic theorem and its appendix proof do not seem fully aligned, and the proof as written does not cleanly establish the stated vanishing-error claim.
3. The real-data evaluation is limited to a single small benchmark.
4. The presentation needs polishing in several places, including citation formatting and some wording in the theoretical sections.

**Additional Comments:**

None.

**Audience:**

Yes

**Audience Explanation:**

I believe the paper would be of interest to readers working on causal discovery, scalable structure learning, and modular learning pipelines.

**Broader Impact Concerns:**

None.

**Claims And Evidence:**

No

**Claims Explanation:**

The submission provides encouraging empirical evidence that the weighted voting version of VISTA can improve precision and runtime across several synthetic settings, and it also shows a positive comparison against DCILP together with modest gains on the Sachs benchmark. So the main practical idea is supported to some extent.

However, I do not think the full set of claims is yet supported convincingly enough. First, the theory is built on an independence assumption for local votes, while the paper itself acknowledges that this assumption is idealized because the local subgraphs overlap and are learned from the same dataset. This makes the current guarantees feel more heuristic than rigorous. Second, to the best of my understanding, the asymptotic consistency statement and the appendix proof appear mismatched: the theorem states one condition on the constant $C$, while the appendix derives a different threshold and then only informally argues that a slightly larger constant yields vanishing error. Third, the empirical validation on real data is narrow, with essentially one small benchmark.

**Requested Changes:**

**Critical:**

1.  The paper should clearly distinguish what is formally proved under explicit assumptions from what is only an intuition for the dependent-vote setting. In particular, the asymptotic consistency theorem and Appendix D.3 should be reconciled carefully, since the statement and derived condition on $C$ do not currently appear to match, and the proof does not yet cleanly establish the stated $o(1)$ conclusion.

2. Since the paper itself acknowledges that the independence assumption is unrealistic, this issue should be treated as a central limitation rather than a minor remark. At minimum, the authors should discuss more concretely how overlap among local subgraphs affects the guarantees, and ideally provide either a weaker but valid guarantee or an empirical study showing the effect of correlated votes.

3. The paper would benefit from additional real or semi-real benchmarks, since the current real-data section only reports Sachs. This is especially important because the method is motivated as a practical large-scale framework.

4. The reported speedups are promising, but the paper attributes them to decomposition and parallel execution of local subgraphs. The authors should report clearly whether timings are serial or parallel wall-clock times, what level of parallelism was used, and whether the same compute budget is being compared across methods.

5. The current results show that naive voting can catastrophically increase false discoveries, and weighted voting can still reduce TPR noticeably for some base learners. These regimes should be discussed more candidly in the main text.

**Would strengthen the paper:**

6. Move the comparison to the closest divide-and-conquer baseline, DCILP, into the main paper or summarize it more prominently, since it is one of the most relevant direct comparisons.

7. Add an ablation on the ordering of cycle removal and thresholding, and on the sensitivity to Markov Blanket quality, since these design choices appear central to the method.

8. The citation format appears incorrect in some places, and the authors should re-check the use of `\citet` versus `\citep` throughout. More generally, the presentation would benefit from another careful proofreading pass.

9. The related work section could be broadened slightly. In addition to the causal-discovery references already discussed above, the authors may also wish to mention relevant work [1], perhaps as a future-work discussion, since learned subgraph selection policies may be conceptually relevant to adaptive neighborhood selection.


**References:**

[1] Efficient Subgraph GNNs by Learning Effective Selection Policies

---

> ### Author Response · Authors · 2026-06-11
> **Reply to Reviewer Z7Xv (Part 1 of 3)**
>
> We thank the reviewer for the careful reading and constructive suggestions. Since many of the concerns raised in the Weaknesses and Discussion sections are also summarized in the `Requested Changes`. Therefore, to avoid repetition, we respond to the reviewer’s `Requested Changes` point by point below and describe the corresponding manuscript revisions.
>
> > The paper should clearly distinguish what is formally proved under explicit assumptions from what is only an intuition for the dependent-vote setting. In particular, the asymptotic consistency theorem and Appendix D.3 should be reconciled carefully, since the statement and derived condition on $C$ do not currently appear to match, and the proof does not yet cleanly establish the stated $O(1)$ conclusion.
>
> Thank you for pointing this out. We have revised the theoretical discussion to more clearly distinguish between what is formally proved under explicit assumptions and what should be interpreted as intuition in the dependent-vote setting.
>
> First, after Theorem 3.2, we now explicitly state that the finite-sample concentration bound is proved under an idealized independence assumption across local subgraph votes. We further explain that, in practice, overlapping Markov Blanket subgraphs can induce correlated votes for the same candidate edge. Such dependence reduces the effective number of independent votes, so the theoretical bound should be interpreted as an idealized benchmark or qualitative guide in the dependent-vote setting, rather than as a formal guarantee for arbitrary correlated votes.
>
> Second, we carefully reconciled the margin used in the proof with the weighted-voting threshold. In particular, the relevant effective threshold is $r_\lambda(m)=\frac{t}{1-e^{-\lambda m}}$, not simply $t$. We therefore explicitly control the gap $r_\lambda(m)-t$, show that it vanishes as $m=C\log n$ grows, and then use this to establish the false-negative bound.
>
> We also revised Appendix D.3 to make the proof of asymptotic consistency more explicit. The proof now decomposes the global error into false-positive and false-negative contributions, accounts for the fact that the number of false candidate edges is $O(n^2)$ while the number of true edges is $O(n)$, and shows that both terms vanish when the number of local votes per candidate edge grows as $m=C\log n$ with a sufficiently large constant $C$.
>
> Finally, we clarified the relationship between the condition on $C$ used in the earlier appendix proof and the condition now stated in the theorem. The previous appendix used the simpler sufficient condition $C>\frac{2}{\min\{\delta_p^2,\delta_q^2\}}$. This condition is stronger than the sharper requirement $C>\max\{\frac{1}{2\delta_p^2},\frac{1}{\delta_q^2}\}$, because $C>\frac{2}{\min{\delta_p^2,\delta_q^2}}$ could induce $C>\max\{\frac{1}{2\delta_p^2},\frac{1}{\delta_q^2}\}$. Thus, the previous condition was not mathematically incorrect, but it was simply a more conservative sufficient condition that implies the sharper one. To avoid any perceived mismatch between the main statement and Appendix D.3, we have now revised both places to use the sharper and more precise form $C>\max{\frac{1}{2\delta_p^2},\frac{1}{\delta_q^2}}$.

---

> ### Author Response · Authors · 2026-06-11
> **Reply to Reviewer Z7Xv (Part 2 of 3)**
>
> > Since the paper itself acknowledges that the independence assumption is unrealistic, this issue should be treated as a central limitation rather than a minor remark. At minimum, the authors should discuss more concretely how overlap among local subgraphs affects the guarantees, and ideally provide either a weaker but valid guarantee or an empirical study showing the effect of correlated votes.
>
> We would like to clarify that this assumption is not required by the VISTA algorithm itself. VISTA only aggregates the directed edge outputs of local subgraph learners, and its decomposition, weighted voting, and acyclicity-enforcement steps remain well-defined regardless of whether the local votes are independent. The independence assumption is used only to derive a clean concentration-style sufficient condition for the weighted-voting guarantee. Therefore, we view this primarily as a limitation of the simplified theoretical analysis rather than a central limitation of our proposed framework. That said, because Markov Blanket subgraphs overlap and are learned from the same dataset, their votes may be positively correlated. This reduces the effective number of independent votes: increasing the raw vote count $m$ may provide less statistical benefit than in the independent case. We have therefore revised the explanation after Theorem 3.2 to explicitly discuss this effect.
>
>
> > The paper would benefit from additional real or semi-real benchmarks, since the current real-data section only reports Sachs. This is especially important because the method is motivated as a practical large-scale framework.
>
>
> Thank you for this helpful suggestion. In addition to the Sachs protein-signaling benchmark, we now include experiments on causalAssembly, a recently proposed semi-synthetic benchmark derived from a real automotive manufacturing line in Section 4.3. This benchmark is much larger than Sachs, containing 98 variables organized into five sequential production stations and 485 directed edges. Its ground-truth DAG is constructed from documented production-process physics, while the observational data are generated from models fitted to real line measurements. We evaluate the same set of base learners both in isolation and when integrated with VISTA, using the same Markov Blanket estimator and the same fixed aggregation parameters $\lambda=0.5$ and $t=0.7$, without any dataset-specific tuning. The new results are reported in Table 6. Overall, VISTA improves the structural recovery performance on this larger semi-real benchmark, generally reducing FDR and SHD and improving F1 across the tested base learners.
>
>
> > The reported speedups are promising, but the paper attributes them to decomposition and parallel execution of local subgraphs. The authors should report clearly whether timings are serial or parallel wall-clock times, what level of parallelism was used, and whether the same compute budget is being compared across methods.
>
> The runtimes reported in Table 3 as well as in 7，8 and 9 are parallel wall-clock times for VISTA. In our implementation, the local Markov Blanket subgraph learning tasks are independent and are executed in parallel. The speedup of VISTA comes from its ability to exploit parallelism across independent local subgraph tasks, whereas the original global learners are run as their standard single global optimization procedures. We have revised the paper and add those details.
>
> > The current results show that naive voting can catastrophically increase false discoveries, and weighted voting can still reduce TPR noticeably for some base learners. These regimes should be discussed more candidly in the main text.
>
> Thanks for the suggestion. We have revised the main text to clarify the intended role of naive voting. VISTA-NV is not proposed as the final recommended estimator. It is rather included as a diagnostic variant to verify that the MB decomposition preserves true causal edges in the candidate pool. Since NV performs no pruning or confidence calibration, it can indeed introduce many spurious edges and substantially increase FDR, as shown in Table 1. The main point of NV is therefore its high TPR, not its precision.
>
> We have also added a detailed discussion of weighted voting. WV is the intended aggregation rule because it down-weights low-support orientations and filters unreliable edges. This often leads to much lower FDR and SHD and better F1, but it can also remove weakly supported true edges and therefore reduce TPR for some base learners. We now explicitly describe this as a precision--recall trade-off rather than claiming uniform improvement across all metrics.

---

> ### Author Response · Authors · 2026-06-11
> **Reply to Reviewer Z7Xv (Part 3 of 3)**
>
> > Move the comparison to the closest divide-and-conquer baseline, DCILP, into the main paper or summarize it more prominently, since it is one of the most relevant direct comparisons.
>
> Thanks for this helpful suggestion. We have moved the comparison with DCILP from the appendix to the main paper as a dedicated subsection, Section 4.2 in the revised manuscript.
>
> > Add an ablation on the ordering of cycle removal and thresholding, and on the sensitivity to Markov Blanket quality, since these design choices appear central to the method.
>
> Thank you for the suggestion. We agree that both the ordering of cycle removal and thresholding, as well as the quality of MB estimation, are important for understanding VISTA. We have therefore expanded the discussion of these two design aspects in the revised manuscript.
>
> For the ordering between GreedyFAS and thresholding, we would like to clarify that this is not treated as a tunable hyperparameter in VISTA, but as a fixed part of the algorithmic design. The purpose of GreedyFAS is to resolve global directional conflicts using the full weighted evidence obtained from local subgraphs. If thresholding is applied first, many low-weight edges are removed before the acyclicity projection. Although this may remove some cycles, any cycles that remain are likely to consist of relatively high-confidence edges. GreedyFAS would then be forced to remove one or more of these stronger edges in order to restore acyclicity, which can severely damage the recovered structure. In contrast, applying GreedyFAS before thresholding allows the algorithm to use all available edge-weight information and preferentially remove weaker conflicting edges; thresholding is then used only as the final sparsification step.
>
> For this reason, we believe that reversing the order would not provide a meaningful ablation of a design choice, but would instead evaluate a deliberately degenerate variant that violates the intended role of the acyclicity projection. Such a result would be difficult to interpret and would not add substantial explanatory value beyond the algorithmic rationale already provided. We have revised the main text to make this rationale more explicit.
>
>
> For the sensitivity to MB quality, we clarify that VISTA is designed to be agnostic to the specific MB identification algorithm. The aggregation and acyclicity steps operate only on the local edge-level outputs and do not rely on the internal assumptions of a particular MB learner. Therefore, an exhaustive ablation over different MB learners would mainly evaluate the chosen MB estimators rather than the proposed aggregation mechanism itself. Nevertheless, MB quality affects the candidate set available to VISTA: if a true edge is excluded from all estimated local neighborhoods, no downstream aggregation rule can recover it. We have added this discussion to clarify that VISTA benefits from high-recall MB estimates, while remaining plug-and-play with respect to the MB identification method.
>
> > The citation format appears incorrect in some places, and the authors should re-check the use of \citet versus \citep throughout. More generally, the presentation would benefit from another careful proofreading pass.
>
> We have carefully double-checked the citation formatting throughout the manuscript and revised the use of textual and parenthetical citations where appropriate.
>
> > The related work section could be broadened slightly. In addition to the causal-discovery references already discussed above, the authors may also wish to mention relevant work [1], perhaps as a future-work discussion, since learned subgraph selection policies may be conceptually relevant to adaptive neighborhood selection.
>
> Thanks for the comment. We have added the recommended paper into our related works.
>
>
> In summary, we appreciate the reviewer’s thoughtful feedback, which helped us strengthen both the theoretical presentation and the experimental evaluation. We hope the revised manuscript is clearer and could address the reviewer’s concerns.

---

### Review · Reviewer_Uzi4 · 2026-06-09

**Summary Of Contributions:**

The manuscript proposed Voting-based Integration of Subgraph Topologies for Acyclicity (VISTA), a plug-and-play and fully parallelizable causal discovery method designed for high-dimensional settings, where standard methods suffer from computational intractibility. The idea is to apply any causal discovery method on all local subgraphs based on Markov Blankets and then apply an exponentially weighted voting mechanism to consolidate the sub-graphs into a global one. Finally, potential cycles in the estimated graph are removed using the Greedy Feedback Arc Set (GreedyFAS) algorithm. VISTA comes with various theoretical guarantees such as asymptotic consistency and a lower bound on weighted voting accuracy. Finally, VISTA is demonstrated on various synthetic and real-world data sets, showing that VISTA typically increases precision without sacrificing recall. At the same time, due to parallelizability, VISTA is much faster to evaluate than standard algorithms like NOTEARS.

**Additional Comments:**

The experiments use $\lambda=0.5$ and $t=0.7$, but why are they chosen like that? Let's say I am given a new causal discovery problem with many nodes. How would I choose $\lambda$ and $t$? I understand how they interplay, but is there some heuristic on how to generally choose them and why? I understand that Theorem 3.4 aims to help here, but the $\lambda$ that we would choose would be different for each edge, because $m$ will generally be different, no? It appears however, that most of the other theory assumes that $\lambda$ is fixed globally.

**Audience:**

Yes

**Audience Explanation:**

The manuscript addresses a relevant problem in a simple, yet principled manner. I think many individuals will be interested in this work.

**Claims And Evidence:**

Yes

**Claims Explanation:**

The method is supported by meaningful theory and empirical evidence. All claims are supported and limitations are properly highlighted. My only minor concern is that there is no comparison with any of the existing "Scalable or Modular Structure Learning" methods described in the related work section. While comparing with these baselines would further strengthen the work, it is not necessary to meet the acceptance criteria (outperforming these methods is not claimed).

**Requested Changes:**

## Section 3

- The notation $\mathcal{G}[S]$ for some set of nodes $S$ is not clear to me. It induces a sub-graph with these nodes, but how would it even be possible that any edges are lost in Proposition 3.1? If all nodes are included in $\mathcal{G}'$ by construction, then so are the edges (?). I suggest formalizing the above mentioned notation.

## Section 3.1

- *"However, while NV does not distinguish between strong and weak statistical support"* This sentence does not appear to be complete. Please fix.

## Figure 3

- I find the figure very helpful and beautiful. Would it be possible to add the labels (1), (2), (3) from the caption to the figure? It would help greatly.

- Another issue is that this figure seems to be at odds with Figure 2. Figure 2 says that the weighted voting is applied before the GreedyFAS post-pruning, while Figure 3 shows that GreedyFAS is applied before the weighted voting filter (if $s < t$, the edge is removed). Which one is true? Please clarify.

## Theorem 3.5

- I cannot see where $global \; error$ is defined. Please clarify.

## Section 4.2

- It is surprising that VISTA is useful for the Sachs data set, given that it is not even so large. How can that be? Given these results, the claims might be broadened a bit even.

---

> ### Author Response · Authors · 2026-06-11
> **Reply to Reviewer Uzi4 (Part 1 of 2)**
>
> We sincerely thank the reviewer for the detailed and helpful comments. We have carefully considered each point and provide our responses below.
>
> > The notation $\mathcal G[S]$ for some set of nodes $S$ is not clear to me. It induces a sub-graph with these nodes, but how would it even be possible that any edges are lost in Proposition 3.1? If all nodes are included in $\mathcal G'$ by construction, then so are the edges (?). I suggest formalizing the above mentioned notation.
>
> Thanks for the comment. We have added the explanation into proposition 3.1 and would like to clarify that $G[S]$ denotes the node-induced subgraph of $G$ on the node set $S$, containing all nodes in $S$ and only those edges of $G$ whose two endpoints both lie in $S$. Thus, even if the union of local subgraphs covers all nodes, it does not automatically cover all edges: an edge is included only when its two endpoints co-occur in at least one local subgraph.
>
> It uses the Markov Blanket property that, for any true edge $X\to Y$, we have $Y\in MB(X)$ and $X\in MB(Y)$. Hence the two endpoints co-occur in the corresponding MB-induced subgraphs, so the edge $X\to Y$ is included in the union. So it serves as a necessary condition for the correctness of our algorithm.
>
> > "However, while NV does not distinguish between strong and weak statistical support" This sentence does not appear to be complete. Please fix.
>
> Thanks for the observation. We have now revised the sentence.
>
> > I find the figure very helpful and beautiful. Would it be possible to add the labels (1), (2), (3) from the caption to the figure? It would help greatly.
>
> We have revised the caption to explicitly label the three stages as Divide, Conquer, and Merge, matching the corresponding modules in the figure and making the workflow easier to follow.
>
> > Another issue is that this figure seems to be at odds with Figure 2. Figure 2 says that the weighted voting is applied before the GreedyFAS post-pruning, while Figure 3 shows that GreedyFAS is applied before the weighted voting filter (if $s<t$, the edge is removed). Which one is true? Please clarify.
>
> Thank you for pointing out this ambiguity. The correct order in VISTA is: first compute the weighted-voting scores, then apply GreedyFAS to remove cycles, and finally apply the threshold filter to remove low-confidence edges. We have revised the pseudocode to separate these steps explicitly: weighted voting now only computes edge scores using $\lambda$, GreedyFAS is then applied to the weighted merged graph, and the threshold $t$ is applied only in the final filtering step. We have now revised the figure 2.
>
> > I cannot see where global error is defined. Please clarify.
>
> We have revised Theorem 3.5 to explicitly define the global error event as the event that the aggregation makes at least one edge-level mistake over all candidate pairs, including either missing a true edge or including a false edge.
>
> > It is surprising that VISTA is useful for the Sachs data set, given that it is not even so large. How can that be? Given these results, the claims might be broadened a bit even.
>
> Thank you for the comment. We agree that the Sachs result suggests that VISTA can also be useful beyond strictly large-scale settings. However, we would prefer not to overstate this point. For a small graph such as Sachs, the benefit of VISTA depends on the behavior of the base learner. If a standalone base learner already performs very well on the full graph, then decomposing the graph into local subgraphs may introduce additional subgraph-level noise and may not provide further improvement. This is precisely why our main claim focuses on large-scale causal discovery. In the Sachs experiment, several standalone base learners are relatively noisy, and VISTA can still help by aggregating local edge-level evidence and filtering low-confidence predictions, thereby reducing false discoveries. Therefore, the Sachs result should be interpreted as evidence that VISTA may improve robustness when base learners are unstable, rather than as a claim that decomposition is always beneficial for small graphs.

---

> ### Author Response · Authors · 2026-06-11
> **Reply to Reviewer Uzi4 (Part 2 of 2)**
>
> > The experiments use $\lambda=0.5$ and $t=0.7$, but why are they chosen like that? Let's say I am given a new causal discovery problem with many nodes. How would I choose $\lambda$ and $t$? I understand how they interplay, but is there some heuristic on how to generally choose them and why? I understand that Theorem 3.4 aims to help here, but the $\lambda$ that we would choose would be different for each edge, because $m$ will generally be different, no? It appears however, that most of the other theory assumes that $\lambda$ is fixed globally.
>
> In our experiments, we use $\lambda=0.5$ and $t=0.7$ as a fixed operating point across all main tables. This choice was selected after preliminary tests as a balanced setting: it is neither too conservative nor too permissive, and it is consistent with the admissible range suggested by Theorem 3.4 for the typical vote counts encountered in our experiments. Intuitively, $\lambda$ controls how quickly the confidence weight $1-e^{-\lambda m}$ increases with the number of supporting subgraphs, while $t$ controls the final acceptance threshold. A larger $\lambda$ reduces the penalty on low-support edges and therefore tends to increase recall, but may also introduce more false positives. A larger $t$ makes the final graph more conservative and usually improves precision, but may lower recall.
>
> And for the parameter chosen, we do not choose a different $\lambda$ for each edge. Instead, $\lambda$ is indeed a global parameter. The edge-specific vote count $m$ is already incorporated into the score through the term $1-e^{-\lambda m}$, so edges with different support levels are automatically calibrated by the same weighting function.
>
> So for a totally new problem without ground truth, one practical strategy is to first compute and cache all local subgraph outputs. The subsequent merging step is lightweight and does not require retraining the base learners, so one can efficiently sweep a small grid of $\lambda$ and $t$ values and choose the operating point according to the desired graph properties. For example, if a user wants a denser, high-recall candidate graph, one may use a larger $\lambda$ or a smaller $t$. If a user wants a cleaner, high-precision graph, one may use a smaller $\lambda$ or a larger $t$.
>
> In summary, we thank the reviewer for the detailed and constructive comments. These suggestions helped us clarify several important points in the presentation. We hope that the revised manuscript and the explanations above could address the reviewer’s concerns.